# Inhibition of type 1 immunity with tofacitinib is associated with marked improvement in longstanding sarcoidosis

William Damsky [1,2✉], Alice Wang[1], Daniel J. Kim[1], Bryan D. Young[3], Katelyn Singh[1], Michael J. Murphy[1], Joseph Daccache [1], Abigale Clark[4], Ruveyda Ayasun [5], Changwan Ryu[6], Meaghan K. McGeary [2], Ian D. Odell[1,7], Ramesh Fazzone-Chettiar[3], Darko Pucar[8], Robert Homer [2], Mridu Gulati[6], Edward J. Miller[3], Marcus Bosenberg [1,2,7], Richard A. Flavell [7,9] & Brett King [1✉]

Sarcoidosis is an idiopathic inflammatory disorder that is commonly treated with gluco-corticoids. An imprecise understanding of the immunologic changes underlying sarcoidosis has limited therapeutic progress. Here in this open-label trial (NCT03910543), 10 patients with cutaneous sarcoidosis are treated with tofacitinib, a Janus kinase inhibitor. The primary outcome is the change in the cutaneous sarcoidosis activity and morphology instrument (CSAMI) activity score after 6 months of treatment. Secondary outcomes included change in internal organ involvement, molecular parameters, and safety. All patients experience improvement in their skin with 6 patients showing a complete response. Improvement in internal organ involvement is also observed. CD4$^+$ T cell-derived IFN-γ is identified as a central cytokine mediator of macrophage activation in sarcoidosis. Additional type 1 cytokines produced by distinct cell types, including IL-6, IL-12, IL-15 and GM-CSF, also associate with pathogenesis. Suppression of the activity of these cytokines, especially IFN-γ, correlates with clinical improvement. Our results thus show that tofacitinib treatment is associated with improved sarcoidosis symptoms, and predominantly acts by inhibiting type 1 immunity.

[1] Department of Dermatology, Yale School of Medicine, New Haven, CT, USA. [2] Department of Pathology, Yale School of Medicine, New Haven, CT, USA. [3] Section of Cardiovascular Medicine, Yale School of Medicine, New Haven, CT, USA. [4] Kansas City University of Medicine and Biosciences, Kansas City, MO, USA. [5] Laura and Isaac Perlmutter Cancer Center, New York University Langone Medical Center, New York, NY, USA. [6] Seciton of Pulmonary, Critical Care, and Sleep Medicine, Yale School of Medicine, New Haven, CT, USA. [7] Department of Immunobiology, Yale School of Medicine, New Haven, CT, USA. [8] Department of Radiology and Biomedical Imaging, Yale School of Medicine, New Haven, CT, USA. [9] Howard Hughes Medical Institute, Yale University School of Medicine, New Haven, CT, USA. ✉email: william.damsky@yale.edu; brett.king@yale.edu

Sarcoidosis is an inflammatory disorder that most commonly affects the lungs, however, any organ including the skin can be involved. Glucocorticoid-based regimens (prednisone and corticotropin gel) are the only approved therapies. Prednisone remains the recommended first-line treatment for both pulmonary and extra-thoracic sarcoidosis[1]. However, this is not ideal as chronic therapy is often required and glucocorticoid-associated toxicities are common[2]. Methotrexate is commonly used as a steroid-sparing agent, but it is often inadequate[1]. TNF inhibitors have also been evaluated; however, the benefit in controlled trials has been marginal, with several studies finding no benefit[3–8]. TNF inhibitors can also induce or exacerbate sarcoidosis[9].

A hallmark of sarcoidosis is the non-caseating granuloma, which is observed microscopically in affected tissues. Although the pathogenesis of granuloma formation in sarcoidosis is incompletely understood, it appears to involve coordinated activity of several cytokines, chemokines, and other signals[10]. While many studies have identified a predominantly type 1 immune response in sarcoidosis (e.g. cytokines including IL-2, IL-12, IL-18, and IFN-γ); others have identified type 3 cytokines (e.g. IL-17 family), or co-production of Type 1 (IFN-γ) and Type 3 (IL-17) cytokines (Th17.1 phenotype)[11–13]. Other studies have even implicated Type 2 cytokines such as IL-4 and IL-13[14,15]. However, the core cytokine drivers of this disease, and those correlating most closely with disease activity and response to therapy, remain imprecisely defined and have hindered therapeutic progress in sarcoidosis.

Many cytokines implicated in sarcoidosis including IFN-γ and IL-2 (Type 1), IL-23 (Type 3), and IL-4 and IL-13 (Type 2) signal via the JAK – signal transducer and activator of transcription (STAT) pathway. Indeed, JAK-STAT pathway activation has been reported in tissues and blood of patients with sarcoidosis[16–20]. JAK inhibitors are oral small molecules that attenuate the activity of JAK-STAT-dependent cytokines. We and others have recently described effective treatment of individual patients with sarcoidosis using tofacitinib (JAK1/3 > 2 inhibitor) or ruxolitinib (JAK1/2 inhibitor); however, this approach has not yet been evaluated in a prospective fashion and the key cytokines targeted remain to be fully elucidated[19–25].

Given the potential promise of JAK inhibition, through inhibition of cytokine activity, in sarcoidosis, here we prospectively evaluate tofacitinib in 10 patients with long-standing cutaneous sarcoidosis (mean duration: 13.2 years), 9 of whom had internal organ involvement. In all 10 patients, disease control with a tofacitinib-based regimen was superior to the preceding immunotherapeutic regimen, particularly for skin involvement. Molecular analyses identify type 1 cytokines (particularly IFN-γ, but also IL-6, IL-12, IL-15, and GM-CSF), but not type 2 or type 3 cytokines, as being activated at baseline and correlating closely with disease activity and response to therapy. Overall, JAK inhibition appears to be effective in sarcoidosis and larger studies to further evaluate this therapeutic approach are warranted.

## Results

**Tofacitinib leads to improvement in cutaneous sarcoidosis.** All 10 patients had cutaneous sarcoidosis. The average age was 56 years (range: 53–63), 4 patients were female, and 6 had brown/black skin (Table 1). The average duration of disease was 13.2 years. Seven patients were being treated for their sarcoidosis at the start of the study, including 5 taking prednisone (Fig. 1a). All patients had moderate-to-severe cutaneous involvement and the average CSAMI activity score, assessed while taking the preceding therapeutic regimen, was 37 (range: 19–56).

All patients had improvement in the CSAMI activity score after 6 months of tofacitinib, with an average reduction of 82.7% (range:

**Table 1 Patient demographics and characteristics.**

| Patient | Age category (years) | Sex | Duration of disease | Prior Tx | Race/ Ethnicity | Histologic pattern in skin[a] | Thoracic (Scadding stage)[b] | Heart | Extra-thoracic LN | Other |
|---|---|---|---|---|---|---|---|---|---|---|
| 1 | 60–69 | F | 6 Yrs | M, P, I | White | Subcut | Yes (2) | Yes | No | None known |
| 2 | 50–59 | F | 6 Yrs | H, M, P | Black/AA | Dermal (including LP) | Yes (1) | No | Yes | Ocular, nasal sinus |
| 3 | 50–59 | M | 5 Yrs | H, M, P | Black/AA | Dermal | Yes (2) | No | Yes | spleen, nasal sinus |
| 4 | 50–59 | M | 16 Yrs | M, P | Black/AA | Dermal (including LP) | Yes (1) | No | No | Nasal sinus |
| 5 | 50–59 | M | 22 Yrs | M, P | White | Dermal and subcut (including LP and dactylitis) | Yes (2) | No | No | Ocular, nasal sinus |
| 6 | 50–59 | F | 27 Yrs | M, P | Black/AA | Dermal (including LP) | Yes (2) | No | No | Laryngeal, nasal sinus |
| 7 | 50–59 | M | 12 Yrs | M, P, I | Black/AA | Dermal and subcut | Yes (1) | No | Yes | None known |
| 8 | 50–59 | M | 6 Yrs | M, A | White | Dermal | Yes (2) | No | Yes | None known |
| 9 | 50–59 | F | 1 Yrs | M, P | White | Dermal | No (0) | No | No | None known |
| 10 | 50–59 | M | 31 Yrs | M, P | Black/AA | Dermal and subcut (including LP) | Yes (2) | No | Yes | None known |

Prior treatments (Tx) encompass all previously utilized therapies; in general trials of at least 3 months (often significantly longer) were attempted. In some cases, prior therapies were discontinued due to adverse effects. The therapeutic regimen for each patient at the beginning of the study is summarized in Fig. 1. Age reported as a range for patient privacy. Race reported according to recent guidelines[58]. AA: African American. H: hydroxychloroquine, P: prednisone, M: methotrexate, I: infliximab, A: azathioprine. LP: signifies the presence of lupus pernio. [a]Dermal: typical dermal-predominant pattern of inflammation histologically, subcut: subcutaneous predominant involvement (Darrier-Roussy pattern). [b]Scadding stage: represents highest documented stage at any point in the patient's history, stage I: adenopathy only, stage 2: adenopathy and parenchymal infiltration.

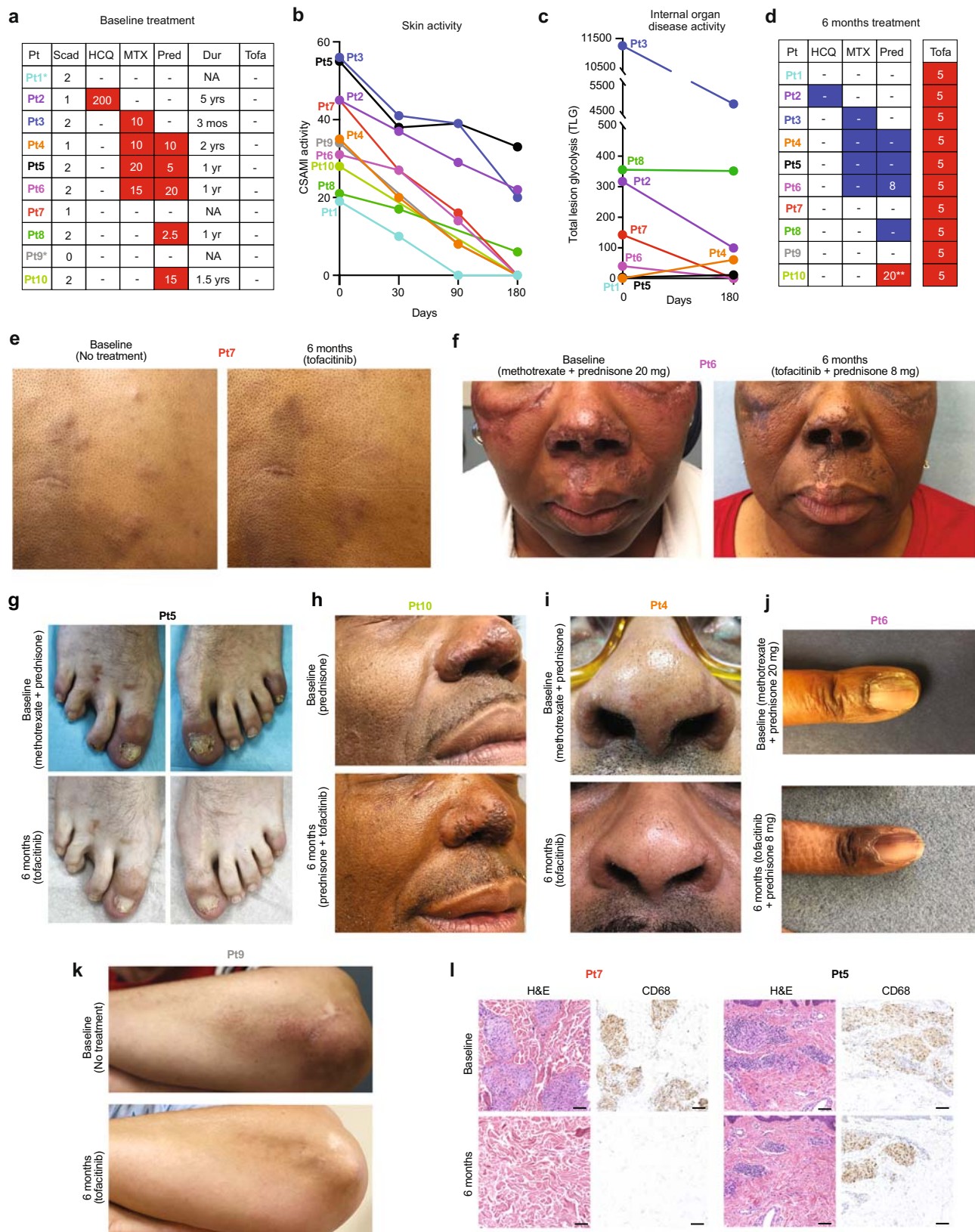

40%–100%) (Fig. 1a–k). Six patients had a compete response (defined as CSAMI activity score of 0), and the remaining four had partial responses (defined as any other decrease in the CSAMI score). Post-inflammatory hyperpigmentation and/or scarring typically persisted (e.g. Fig. 1e–k. All 10 patients reported improved skin-related quality of life (Supplementary Fig. 1).

A skin biopsy was performed in a subset of patients before and after 6 months of treatment. In patients that continued to have active skin lesions while on tofacitinib, a lesion that still appeared active (erythematous, indurated) was biopsied (Pts 2, 5, and 8). In patients with a complete clinical response to tofacitinib, a sample from a previously active area of skin, analogous to the initial

**Fig. 1 Tofacitinib treatment leads to improvement in cutaneous sarcoidosis. a** Left panel: baseline treatment regimens for each patient, HCQ: hydroxychloroquine (dose shown as mg twice daily), MTX: methotrexate (mg weekly), Pred: prednisone (mg daily), and Tofa: tofacitinib, Dur: duration of most recent therapeutic regimen; patients taking prednisone often had been on and off for significantly longer than indicated. *Treatment with prednisone or methotrexate was recommended but declined by patients due to prior adverse effects with these medications. Scad: maximum previous Scadding stage. **b** CSAMI activity scores and **c** total lesion glycolysis (TLG) over the study period. **d** Treatment regimens at the end of the study (6 months), tofacitinib (Tofa) dose shown as mg twice daily, blue: discontinued/reduced dose during study, **prednisone increased due to worsening pre-existing Achilles tendinopathy, not worsening of sarcoidosis. **e** Dermal papules/plaques and subcutaneous nodules of sarcoidosis before and after treatment. **f** Extensive involvement of the face before and after 6 months of treatment; scarring from the longstanding (>20 years) lesions persisted. **g** Sarcoid dactylitis before and after 6 months of treatment. **h** Lupus pernio presentation of sarcoidosis, also with annular plaque on the cheek, before and after treatment; significant scarring persisted in this patient. **i** Lupus pernio presentation of sarcoidosis before and after treatment. **j** Nail dystrophy related to sarcoidal inflammation in the nail matrix (demonstrated by matrix biopsy) resolved after 6 months of treatment. Post-inflammatory hyperpigmentation persisted. **k** Papules and plaques on the forearm of a patient before and after treatment. **l** Hematoxylin and eosin (H&E) stained and CD68 immunohistochemistry on skin biopsies from a representative complete responder (Pt 7, both from the back) and partial responder (Pt 5, both from the arm), scale bar: 150 µM. Similar results were seen in two additional partial responders in whom biopsies were performed. Source data are provided as a Data Source File.

biopsy site, was taken (Pt 7 and two previously published patients[19,20]). Complete clinical response correlated with histologic dissolution of granulomas in the skin, consistent with our prior observations[19,20]. In patients with a partial clinical response, biopsies showed persistence of granulomatous inflammation in the areas that remained clinically active (Fig. 1l).

**Tofacitinib leads to improvement in pulmonary sarcoidosis.** Of the 10 patients, 8 had active pulmonary involvement and 1 had active myocardial involvement (Table 1 and Supplementary Table 1). Whole body PET-CT was performed in these 9 patients and total lesion glycolysis (TLG) was determined to assess the degree of active internal organ sarcoidosis. Baseline studies were performed while the patients were taking their stable preceding immunosuppressive regimen that included prednisone plus methotrexate in 3 patients, and prednisone or methotrexate monotherapy in 2 and 1 patients, respectively (Fig. 1a). A subsequent PET-CT after 6 months of tofacitinib was obtained. One follow-up PET-CT study (of 9) could not be interpreted due to nonadherence to the dietary preparation and so data from 8 patients were analyzed.

A decrease in TLG of ≥50% was noted in 5 patients, with complete or near complete resolution in 3 (defined retrospectively as >98% reduction in TLG) (Fig. 1c and Fig. 2a, b). Improvement was seen in both lymphadenopathy and in parenchymal lung involvement. Another patient had essentially stable TLG, but did appear to have improvement in mild lymph node avidity below the detection threshold (Fig. 2c). Two patients had a slight increase in TLG, however; both were able to discontinue methotrexate and prednisone during the study and the change did not appear to be clinically significant (Fig. 2d). Further, both patients experienced improvement in their skin. One patient had myocardial activity seen only on the dedicated cardiac scan; this activity resolved entirely with therapy (Fig. 2e).

Of the 5 patients taking prednisone at baseline, three were able to discontinue it and one was able substantially decrease the dose. One patient with pre-existing Achilles tendinopathy responsive to prednisone increased his prednisone dose during the study for this issue. All patients successfully discontinued methotrexate. Several patients reported subjective improvement in confirmed or suspected nasal sinus involvement. A patient with a hoarse voice for >10 years due to laryngeal involvement noted normalization of her voice.

In general, the skin tended to improve to a greater degree than internal organ involvement. There was some observed correlation between the degree of improvement in cutaneous and internal organ involvement (Fig. 2f). Definitive interpretation of this trend was limited by varying baseline treatment regimens. Tofacitinib

was well tolerated, and there were no significant or dose-limiting adverse events (Supplementary Table 2). In summary, the treatment regimen including tofacitinib led to disease control that had not been achieved with prior treatment in all 10 patients.

**IFN-γ from CD4[+] T cells activates macrophages.** Many cytokines have been implicated in sarcoidosis and Th1, Th2, and Th17 polarization have all been described[26]. All of these pathways can activate JAK-STAT signaling. In order to better understand granuloma composition and the potential molecular targets of tofacitinib in sarcoidosis, single cell RNA sequencing (scRNA-seq) was performed on lesional skin from 3 of the patients prior to treatment with tofacitinib (Pts 1,2,5) and compared with normal skin from 3 healthy controls (Supplementary Table 3). Although Pt 2 (hydroxychloroquine) and Pt 5 (methotrexate/prednisone) were on other treatments at the time of biopsy; their regimen had been stable and their cutaneous sarcoidosis was still active. A total of 24034 cells were analyzed with a median of 4029 unique molecular identifiers. The clusters were visualized using uniform manifold approximation and projection (UMAP). We found 37 clusters corresponding to 11 major cell types (Fig. 3a–c and Supplementary Fig. 2). Although transcriptional variation was noted in several cell types, when comparing sarcoidosis to normal, we initially focused our analysis on T cells and myeloid cells given the important role that these cell types play in granuloma formation[27].

A total of 11236 T cells were clustered independently of the other cell types, revealing 14 unique clusters composed of CD4[+]FOXP3[-] T helper cells, regulatory T cells (CD4[+]FOXP3[+]), and CD8[+] T cells (CD8A[+]) (Fig. 4a and Supplementary Fig. 3). CD4[+]FOXP3[-] cells were the most abundant T cell type in sarcoidosis, accounting for 70.2% of T cells. CD8[+] T cells accounted for only 14% of all T cells and differences in gene expression patterns were difficult to assess given the paucity of CD8[+] T cells in the healthy controls (Supplementary Table 4).

Clusters 2, 7, and 12 (referred to as "CD4[+] SAR-1") were highly enriched CD4[+]FOXP3[-] clusters in the sarcoidosis samples (Fig. 4b and Supplementary Fig. 3). These cells showed significant cytokine and chemokine production and expressed T resident memory markers ($T_{rm}$) such as ITGAE (CD103), CD69, and ITGA1 (CD49A), as well as cytotoxic markers such as GZMA and GZMB. One cluster was predominant in each patient, but the characteristics were similar. A second population of CD4[+]FOXP3[-] cells was also enriched in the sarcoidosis samples (clusters 3, 8, and 4; referred to as "CD4[+] SAR-2"). These clusters expressed lymphoid homing markers such as CCR7 (Supplementary Fig. 3), consistent with a recirculating population[28]. Significant cytokine expression was not observed in these cells.

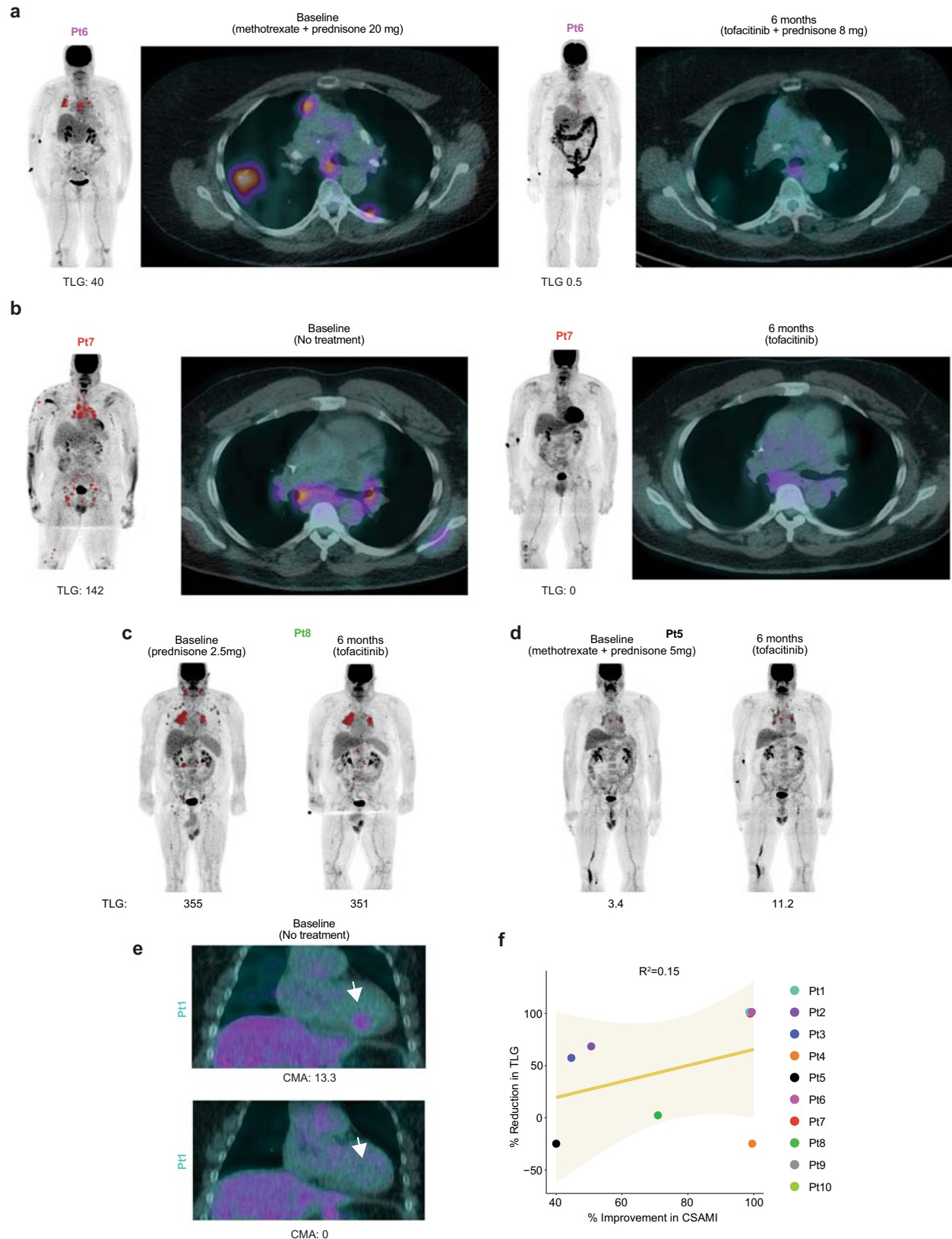

Differential gene expression and pathway (IPA) analysis was used to compare the CD4$^+$ SAR-1 clusters to the predominant CD4$^+$FOXP3$^-$ clusters from controls (CD4$^+$ CTRL clusters: 0, 5) (Fig. 4c, 3d). Evidence of chronic antigenic stimulation was present in the sarcoidosis T cells (Fig. 4e and Supplementary Fig. 3). *IFNG* (IFN-γ) was by far the most differentially

upregulated cytokine in CD4$^+$ SAR-1 clusters, which also expressed high levels of canonical T$_h$1 transcription factors (*TBX21*, *STAT1*). Increased expression of *CSF2* (GM-CSF) was also identified. There was minimal expression of Th2 markers (eg. *IL13*, *IL4*, and *GATA3*) and almost no detectable expression of Th17 markers (e.g. *IL17A*, *IL17F*, *RORC*) (Fig. 4d and

**Fig. 2 Tofacitinib treatment leads to improvement in pulmonary and myocardial sarcoidosis. a, b** PET (coronal) and PET-CT (axial) studies before and after 6 months of tofacitinib in patients with complete or near complete internal organ response, TLG: total lesional glycolysis. **c** PET (coronal) studies before and after 6 months of treatment in a patient with improvement in lymph node avidity below the threshold of quantification. **d** PET (coronal) studies before and after 6 months of treatment in a patient with a slight increase in PET avidity which was felt to be clinically insignificant. **e** Cardiac PET-CT (coronal) before and after 6 months of treatment in a patient myocardial involvement of the inferior intraventricular septum (arrow), CMA: cardiac metabolic activity. **f** Scatterplot comparing change in cutaneous sarcoidosis and extra-cutaneous sarcoidosis. Depicted as simple linear regression line with 95% confidence interval (shared area), Goodness of Fit represented by R squared. Cutaneous involvement shown as percent reduction in CSAMI during the study period. Extracutaneous involvement shown as percent reduction in TLG during the study period; for patients with increase in TLG during the study, worsening of 25% was arbitrarily assigned. Source data are provided as a Data Source File.

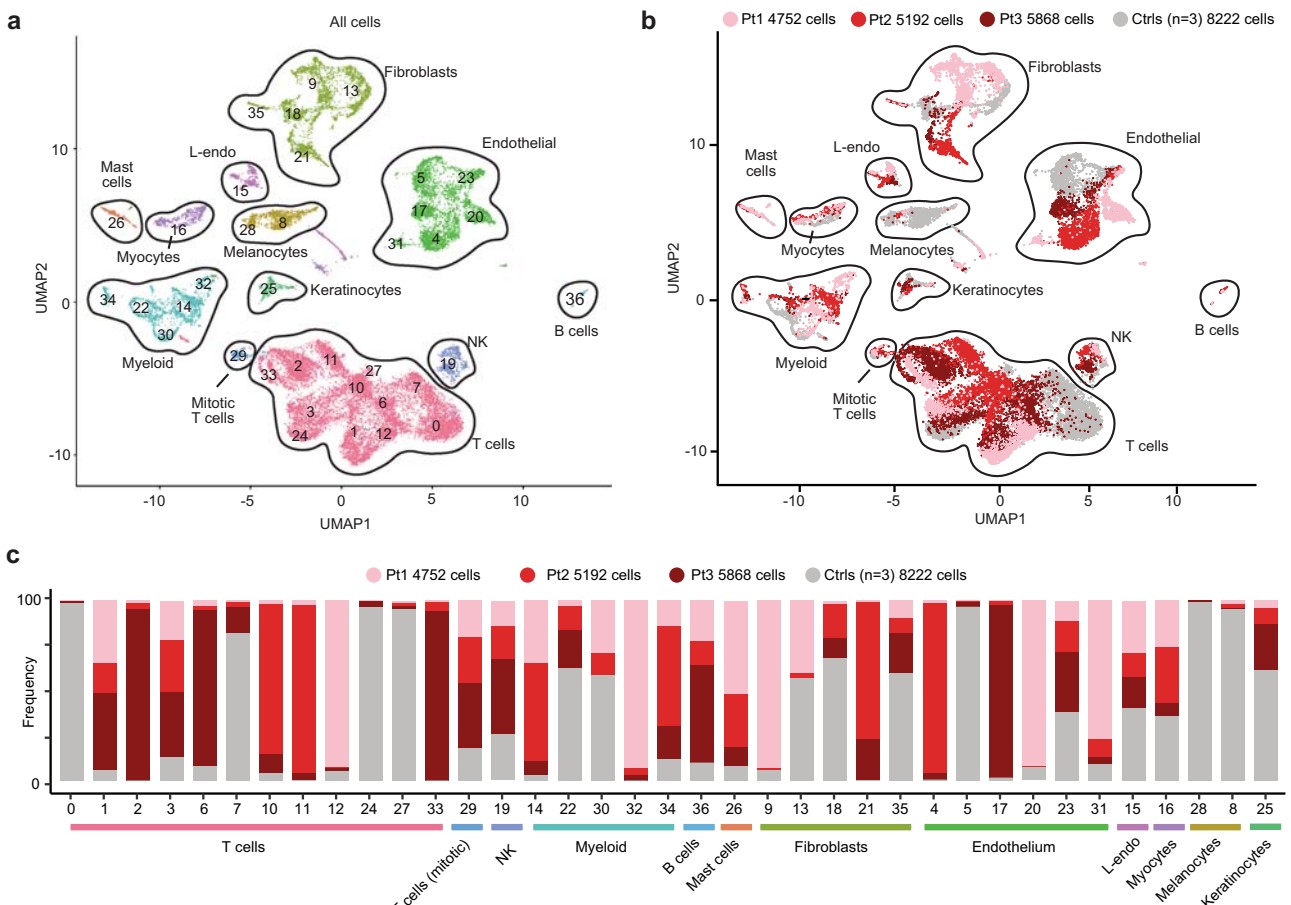

**Fig. 3 scRNAseq of cutaneous sarcoidosis and control skin samples. a** UMAP projection of scRNA-seq data showing clustering of all cells, colored by cell type. **b** UMAP projection of scRNA-seq data in (**a**), colored by condition/library. **c** Histograms showing the contribution of each library/condition to each cluster. NK natural killer cell, L-endo lymphatic endothelium.

Supplementary Fig. 3). *TNF* was expressed at modest levels. Chemokines CCL3, CCL4, and CCL5 were also highly upregulated in CD4$^+$ SAR-1 clusters (Fig. 4c, d).

We next separately analyzed myeloid clusters. A total of 1748 myeloid cells were analyzed, revealing 14 clusters which included both macrophages and dendritic cells (DCs). Macrophages were abundant and somewhat diverse in the sarcoidosis libraries (Fig. 4f–g and Supplementary Fig. 4). Expression patterns among predominant macrophage populations in sarcoidosis (clusters 0, 1, 4, 6, 9, 10; termed "Mac SAR-1") were compared to control myeloid clusters (2, 8, 13). Mac SAR-1 clusters appeared to be predominantly monocyte-derived, expressed high levels of transcription factors *STAT1* and *CEBPB*, activation/effector molecules (*CHIT1*, *CHI3L1*, *TREM1*, *TREM2*), T cell chemokines (*CXCL9*, *CXCL10*, *CXCL11*), and other interferon responsive

genes (*GBP1*, *GBP4*, *GBP5*), among others (Fig. 4h, i). Expression of these transcripts varied somewhat among the clusters enriched in the sarcoidosis samples (Supplementary Fig. 4).

Pathway analysis was performed and indeed IFN-γ was the most highly significant predicted upstream cytokine regulator of Mac SAR-1 clusters based on their transcriptional profile (Fig. 4j) and consistent with the upregulation of *IFNG* observed in CD4$^+$ T cells in sarcoidosis. A second group of macrophages termed MAC-SAR-2 (clusters 3, 7, 11) were also enriched in sarcoidosis, but did not appear to be as clearly responding to IFN-γ. A population of activated DCs that produced *IL12B* were present almost exclusively in the sarcoidosis samples (Supplementary Fig. 4). This production of IL-12 (a JAK-STAT dependent cytokine) is also consistent with a type 1 polarized immune response; this population is discussed further below.

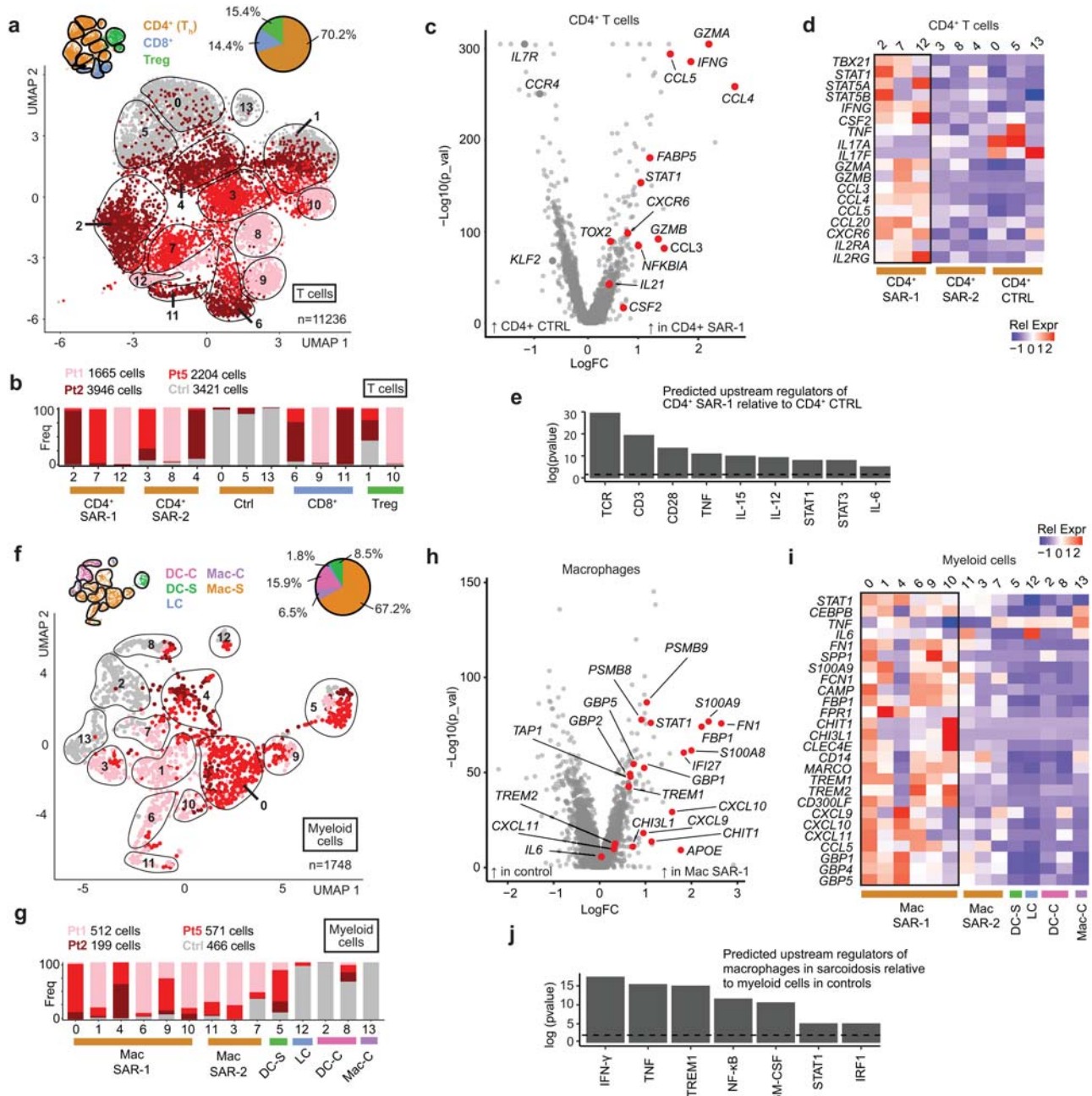

**Fig. 4 IFN-γ produced by Th1 polarized CD4⁺ T cells activates macrophages in cutaneous sarcoidosis. a** UMAP projection of scRNA-seq data showing T cell clusters in sarcoidosis (shades of red) compared to healthy controls (grey). **b** Histograms showing contribution of each condition (sarcoidosis: shades of red, grey: controls) to each T cell cluster. **c** Volcano plot showing the most differentially expressed genes between CD4⁺ SAR-1 (clusters 2,7,12) versus CD4⁺ CTRL (clusters 0,5), corresponding to Fig. 4a. *p* value determined using Wilcon Rank-Sum test, two-tailed. **d** Heatmap showing expression of selected transcripts in *CD4⁺FOXP3⁻* T cell clusters. **e** Histogram showing selected predicted upstream regulators of CD4⁺ SAR-1 clusters (2,7,12) versus CD4⁺ CTRL clusters (0, 5) as determined by IPA. Significance cutoff of *p* < 0.001 is shown by a dotted horizontal line and determined using Fisher exact test, right-tailed. **f** UMAP projection of scRNA-seq data showing myeloid cell clusters in sarcoidosis (shades of red) compared to healthy controls (grey). **g** Histograms showing contribution of each condition (sarcoidosis: shades of red, grey: controls) to each myeloid cluster. **h** Volcano plot showing the most differentially expressed genes between Mac SAR-1 (clusters 0,1,4,6,9,10) versus control myeloid (clusters 2,8,13). *p* value determined using Wilcon Rank-Sum test, two-tailed. **i** Heatmap showing expression of selected transcripts in myeloid clusters. **j** Histogram showing selected predicted upstream regulators in Mac SAR-1 and Mac SAR-2 (clusters 0,1,3,4,6,7,9,10,11) versus myeloid cells in controls (clusters 2,8,13) as determined by IPA. Significance cutoff of *p* < 0.001 is shown by a dotted horizontal line, as determined using Fisher exact test, right-tailed.

**IFN-γ-activation is conserved across sarcoidosis**. Together, these data strongly implicated IFN-γ derived from an aberrant Tₕ1 response as a key driver of sarcoidosis in skin. Next, we evaluated whether a similar population of CD4⁺ T cells could also be identified in pulmonary sarcoidosis. We analyzed scRNAseq data from bronchoalveolar lavage (BAL) fluid from 4 patients with sarcoidosis[29] and compared to 10 healthy control patients[30]. Of 55,908 total cells, 4,621 were T cells (Fig. 5a–e). Although recovery of T cells in the sarcoidosis samples appeared suboptimal, we observed that similarly to the skin, in BAL the

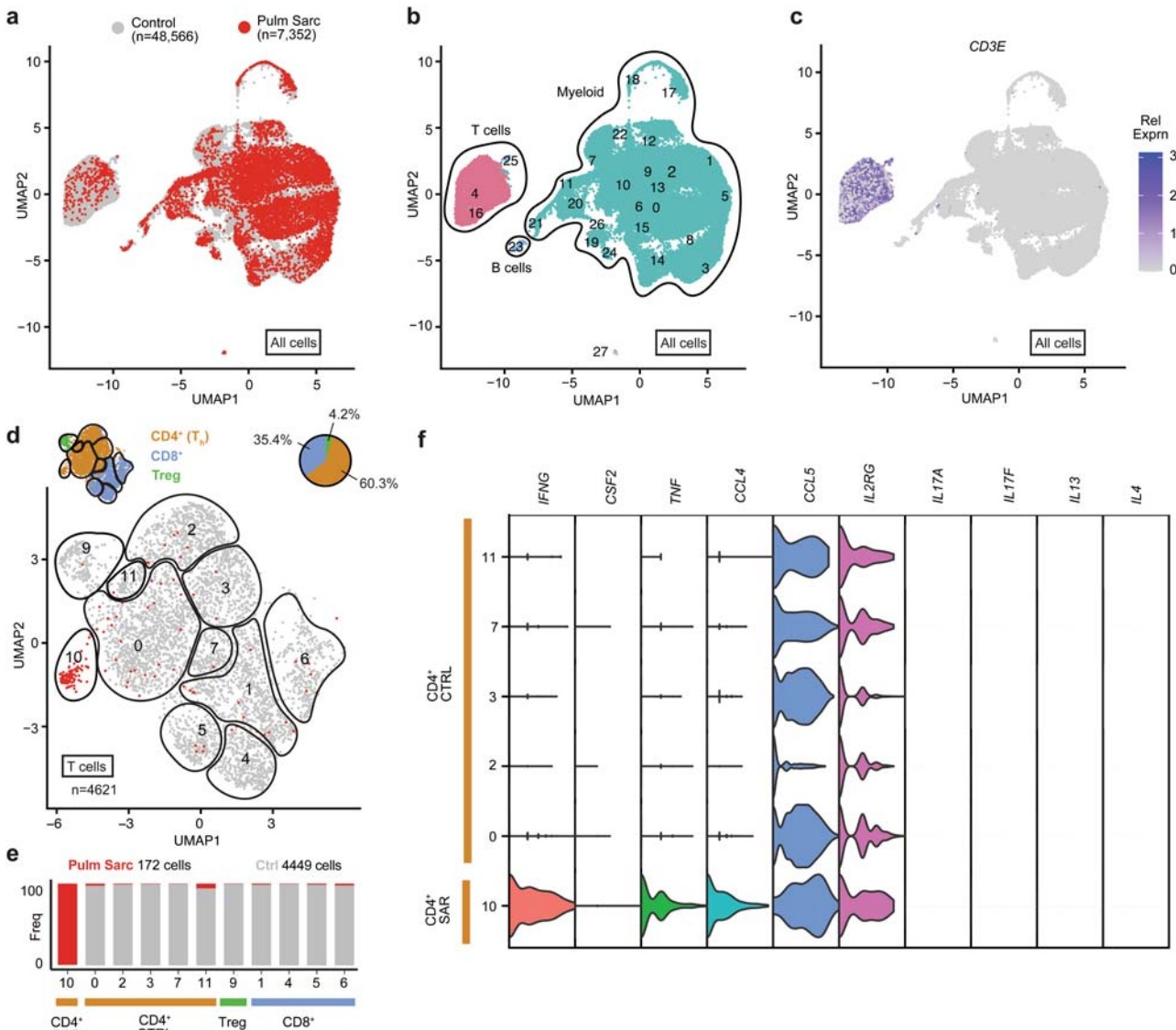

**Fig. 5 Analysis of scRNAseq data of bronchoalveolar lavage (BAL).** Patients with sarcoidosis (n = 4)[29] and healthy control patients (n = 10)[30] were included. **a** UMAP projection of scRNA-seq data of all cells colored by condition (red: sarcoidosis libraries, grey: control libraries). **b** UMAP projection of scRNA-seq data of all cells colored by cell type. **c** UMAP projection of scRNAseq data of all cells showing relative *CD3E* expression. **d** UMAP projection of scRNAseq data of T cell clusters colored by condition (red: sarcoidosis libraries, grey: control libraries). **e** Histogram showing the contribution of each condition (sarcoidosis: red, grey: controls) to each T cell cluster. **f** Violin plots showing expression of select genes in each cluster of *CD4⁺FOXP3⁻* cluster of T cells.

majority of T cells were *CD4⁺FOXP3⁻*. Indeed, a distinct *CD4⁺FOXP3⁻* cluster present exclusively in the sarcoidosis samples, with a Th1 phenotype marked by upregulation of *IFNG* and to a lesser degree *CSF2* (GM-CSF), was identified. Expression of *IL17A*, *IL17F*, *IL4*, and *IL13* by these cells was undetectable (Fig. 5f).

Next, bulk RNA sequencing was performed on skin biopsies from 6 sarcoidosis patients treated with tofacitinib from whom tissue was available and compared to 6 healthy controls (Supplementary Table 5). An independent, previously published gene expression data set consisting of an additional 15 cutaneous sarcoidosis biopsies and 5 controls was also analyzed[31]. IPA was performed to compare gene expression in sarcoidosis versus controls in both data sets. The results were plotted and showed that an IFN-γ transcriptional signature was the most significant change in both data sets; a STAT1 gene expression signature was also highly significant. Additional cytokines including IL-15, IL-6,

GM-CSF, and TNF (and NF-κB) were also among the most significant predicted cytokine regulators of transcriptional patterns in both data sets (Fig. 6a) and consistent with the scRNA-seq data.

Pulmonary sarcoidosis is known to have varied clinical presentations and disease natural history. For this reason, we next analyzed bulk gene expression data from BAL samples from a large cohort of pulmonary sarcoidosis patients with diverse clinical phenotypes[32]. In particular, we analyzed data from 76 untreated sarcoidosis patients with Scadding stage 1 (n = 24), stage 2/3 (n = 40), and stage 4 (n = 12) disease. These data were compared to gene expression data from BAL from 6 healthy controls[33]. Similarly, to the skin, this analysis showed a prominent IFN-γ-signature, both in terms of cytokine production and downstream gene expression (Fig. 6b and Supplementary Fig. 5). The magnitude of the IFN-γ expression (and expression of downstream transcriptional target genes, e.g. *STAT1*, *CXCL9*,

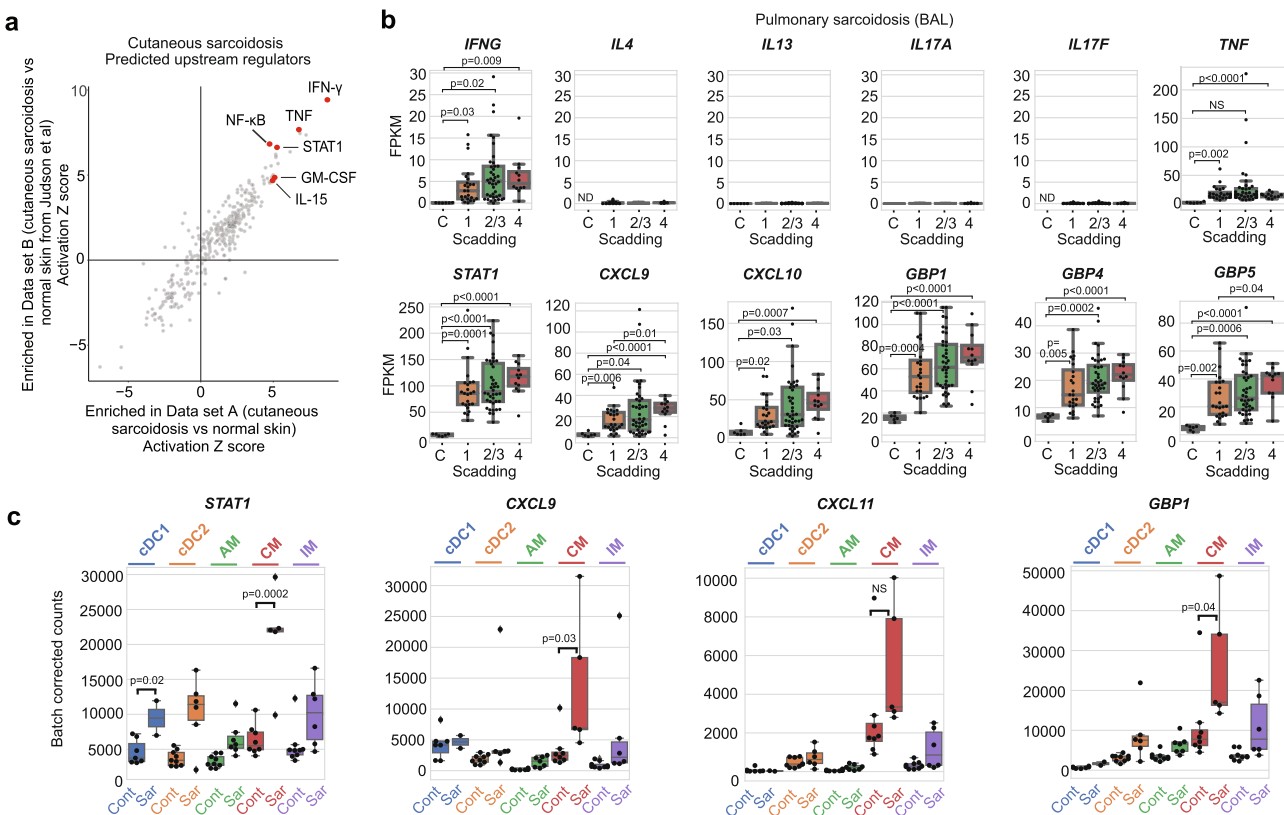

**Fig. 6 IFN-γ signaling is a hallmark of sarcoidosis. a** IPA analysis showing predicted upstream regulators in cutaneous sarcoidosis dataset A (bulk RNAseq from this study) and cutaneous sarcoidosis dataset B (Judson et al);[31] IPA content version: 51963813). Each data set consisted of a series of cutaneous sarcoidosis biopsies and normal control skin. **b** Box and whisker plots showing expression of selected genes in pulmonary sarcoidosis with Scadding stage 1 ($n = 24$), stage 2/3 ($n = 40$), stage 4 ($n = 12$) relative to healthy controls (C) ($n = 6$)[33]. Box plots indicate median (middle line), 25th, 75th percentile (box), and 5th and 95th percentile (whiskers), $p$ values calculated using unpaired $t$ tests, NS: not significant, ND: not determined (abundance below threshold of detection), FPKM: fragments per kilobase of exon per million mapped fragments. See also Supplementary Fig. 5. **c** Box and whisker plots showing expression of selected genes in various populations of FACS purified myeloid cells from BAL of healthy controls ($n = 9$) (Cont) vs sarcoidosis ($n = 8$) (Sar)[34]. Box plots indicate median (middle line), 25th, 75th percentile (box), and 5th and 95th percentile (whiskers), $p$ values calculated using unpaired $t$ tests, NS: not significant, cDC1 and 2: classical dendritic cell types 1 and 2, AM: alveolar macrophage, CM: classical monocyte, IM: intermediate monocyte. (See also Supplementary Fig. 6). Source data are provided as a Data Source File.

*CXCL10*, *GBP1*, GBP4, and *GBP*5) trended towards higher levels with increasing stage (although this was generally not statistically significant); in contrast Th2 and Th17 cytokine expression was negligible in these samples, even in patients with fibrotic disease (e.g. Scadding stage 4) (Fig. 6b).

To assess which myeloid populations in lung might be responding to IFN-γ, we analyzed another previously published data set[34], comparing gene expression patterns among FACS purified myeloid populations from sarcoidosis BAL ($n = 8$) to control BAL ($n = 9$). We found that the expression of IFN-γ response genes was highest in classical monocytes, although alveolar macrophages and other cell types also appeared to be responding to IFN-γ (Fig. 6c and Supplementary Fig. 6).

As BAL might not fully reflect granulomas in the lung parenchyma, we also studied immunologic patterns in tissue containing parenchymal pulmonary granulomas from sarcoidosis patients ($n = 10$) and compared them to cutaneous sarcoidosis ($n = 10$) and controls (Supplementary Table 6). We utilized RNA in situ hybridization to stain for key cytokine markers of immune polarization including Th1 (*IFNG*), Th2 (*IL13*), and Th17 (*IL17A*)[35]. We also included cutaneous inflammatory disorders with canonical Th2 (atopic dermatitis) and Th17 (psoriasis) polarization as positive controls to contextualize patterns seen in sarcoidosis[35]. We found significant expression of *IFNG* in lymphocytes in the sarcoidosis cases, but not normal lung or

skin (Fig. 7a, b). Occasional, weak *IL13* expression was observed in the sarcoidosis cases, but it was not significantly different from controls. No *IL17A* staining was detected in the sarcoidosis cases (Fig. 7a, b). Taken together, the immunologic pattern in the pulmonary granulomas was highly consistent with that seen in the skin samples.

**Other JAK-dependent cytokines play a reinforcing role**. JAK inhibitors can target multiple cytokines simultaneously. In order to better understand how JAK inhibition might control inflammation in sarcoidosis beyond inhibition of IFN-γ, we returned to the IPA analysis comparing CD4+ SAR-1 cells to CD4+ CTRL cells. This revealed that the most significant predicted upstream regulators of this T cell population included IL-6, IL-12, and IL-15 (Fig. 4e). In order to follow-up on this, we examined the skin scRNA-seq data to ask what cell types were producing these cytokines.

We found that the primary source of IL-12 expression was dendritic cells with a cDC1 phenotype (*CLEC9A*+, *IRF8*+, *BATF3*+) (Supplementary Fig. 4). To follow-up on this, we also compared gene expression in the FACS purified myeloid populations from BAL from patients with sarcoidosis ($n = 8$) and controls ($n = 9$)[34]. We indeed found that essentially all of the *IL12B* expression was from a similar population of *IRF8*+*XCR1*+-*BATF3*+ cDC1s in pulmonary sarcoidosis (Supplementary Fig. 6).

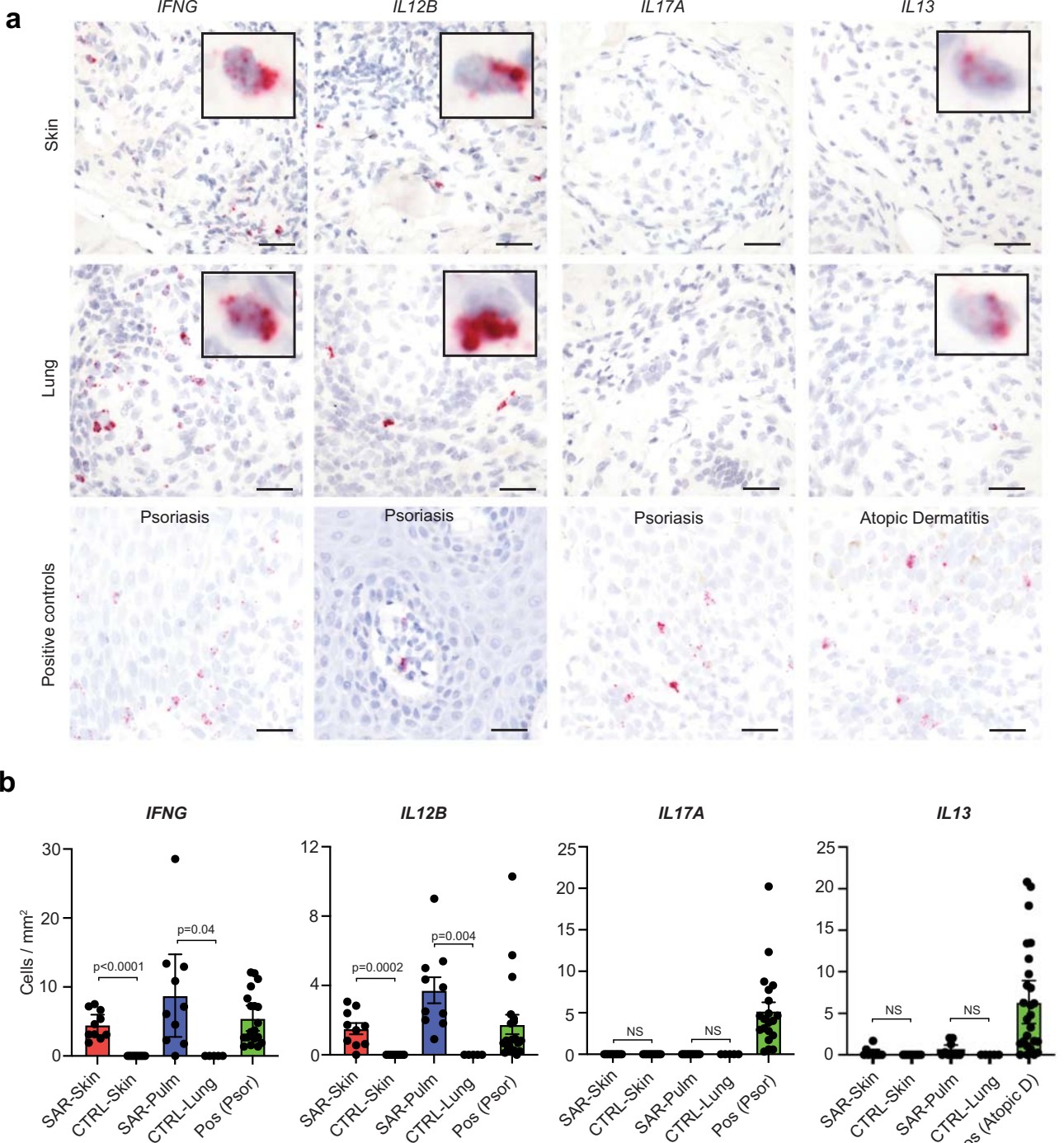

**Fig. 7 Type 1 immunity is the predominant immune polarization in sarcoidosis. a** Photomicrographs showing representative RNA in situ hybridization staining patterns for selected markers (red chromogen) with hematoxylin counterstain (blue), scale bar: 50 μM, higher power inset. **b** Histograms showing quantification of RNA in situ hybridization staining for selected markers in control skin ($n = 10$), control lung ($n = 5$), cutaneous sarcoidosis ($n = 10$) and pulmonary sarcoidosis ($n = 10$) tissue. Data are presented as mean $+/-$ 95% confidence interval, $p$ values calculated using unpaired $t$ tests, NS: not significant, Pos: positive control, Psor: psoriasis biopsies ($n = 20$), Atopic D: Atopic dermatitis biopsies ($n = 26$). Source data are provided as a Data Source File.

RNA in situ hybridization in pulmonary and cutaneous sarcoidosis biopsies also confirmed significant *IL12B* staining in myeloid cells (histiocyte morphology) in both cutaneous and pulmonary sarcoidosis relative to controls (Fig. 7a, b).

We again turned back to the cutaneous scRNA-seq data to look at *IL15* and *IL6* expression. We found that the majority of

expression of these cytokines was from stromal cells: fibroblasts and endothelial cells (Fig. 8a and Supplementary Fig. 7). Fibroblasts in sarcoidosis clustered separately from controls and appeared to primarily be responding to IFN-γ. Sarcoidosis fibroblasts also produced increased levels of chemoattractants (*CCL2, CCL4, CCL5, CXCL9*), suggesting a role in both

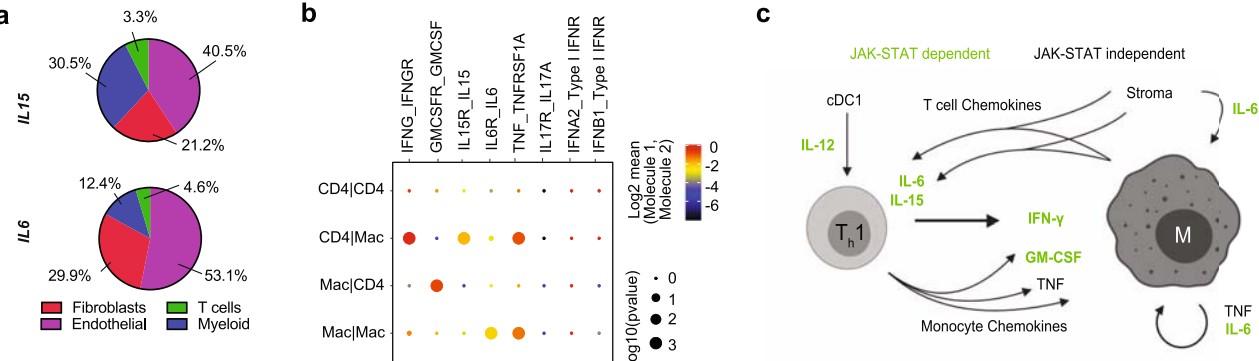

**Fig. 8 Cell type specific receptor-ligand analysis and summary in sarcoidosis. a** Pie chart showing relative proportions of *IL15* and *IL6* producing cells in sarcoidosis scRNA-seq data from skin (see also Supplementary Fig. 7). **b** Dot plot of cellphone DB receptor-ligand interaction analysis for select receptor-ligand pairs, *p* value as determined using Cellphone DB script. **c** Summary of central cytokine and chemokine signals in sarcoidosis revealed by scRNA-seq and other experiments.

maintaining T cell activation as well as in recruiting monocytes and T cells to the tissue microenvironment (Supplementary Fig. 8).

Key proposed cytokine interactions between T cells and macrophages were further supported by an unbiased in silico analysis of receptor-ligand interactions in the cutaneous scRNA-seq data (Cellphone DB)[36] (Fig. 8b). We hypothesize that IL-12, IL-15 and IL-6 play a secondary/reinforcing role to maintain CD4+ T cell recruitment and activation (e.g. production of IFN-γ and to a lesser extent GM-CSF). We hypothesize that inhibition of these multiple cytokines by tofacitinib might explain clinical improvement in sarcoidosis patients (summarized in Fig. 8c).

**Clinical improvement correlates with suppression of IFN-γ.** Given the apparent similarity of molecular drivers among sarcoidosis cases, we next sought to understand why some patients responded better to tofacitinib than others. First, we used bulk RNA-seq to analyze pre- and during-treatment skin biopsies from patients achieving a complete response[19,20] (CSAMI 0 on therapy) versus those exhibiting only partial cutaneous improvement (Supplementary Table 5). CD4+ SAR-1 and Mac SAR-1 upregulated genes (from scRNA-seq experiments; as in Figs. 4d, and 4i) and an IFN-γ transcriptional response signature tended to normalize to levels comparable with healthy controls in complete responders (Fig. 9a). Whereas, in patients with partial improvement, these transcripts were detected at comparable levels before and during treatment. GM-CSF, IL-15, IL-6, and TNF transcriptional signatures also correlated with response patterns, though not as closely as the IFN-γ signature (Supplementary Fig. 9). There were no discernably significant differences in expression of these gene sets between responders and nonresponders at baseline (Fig. 9a).

**Tofacitinib reduces inflammatory markers in plasma.** In order to better understand clinical response patterns at a systemic level, we next turned to a semiquantitative high-throughput protein profiling assay to study plasma from the 9 trial patients with internal organ involvement in whom matched samples were available before and after 6 months of tofacitinib and compared these with 11 healthy controls (Supplementary Table 7). Of the 1536 proteins analyzed with this approach, we found IFN-γ, IL-6, and TNF to be among the most upregulated in sarcoidosis samples at baseline relative to controls (Fig. 9b). IL-15 and IL-12p40 were more modestly increased (GM-CSF was not part of this panel). Levels of Th17 (IL17-A, IL17-F) and Th2 (IL-4, IL-13) cytokines tended to be less abundant in sarcoidosis patients relative to

controls. Further, we found a general correlation between proteins differentially abundant in plasma and differentially expressed in cutaneous sarcoidosis, suggesting upregulation of these markers in plasma was a direct reflection of disease activity (Fig. 9c).

Given the similarity in the pattern of baseline immune dysregulation between complete and partial responders, we wondered if the magnitude of the dysregulation might play a role in the degree of improvement observed with therapy. To examine this, we overlaid 1) proteins upregulated in sarcoidosis plasma at baseline versus controls with 2) proteins modulated by tofacitinib therapy in sarcoidosis plasma, and in doing so revealed a 14-protein plasma signature of sarcoidosis that we termed, "sarcoidosis plasma signature" (SPS). The SPS included IFN-γ, IFN-γ targets (CXCL9, CXCL10, CXCL11), macrophage activation proteins (CHIT1, CHI3L1, and FBP1) and TNF (Fig. 9d). We then evaluated whether differences in SPS profiles correlated with response to tofacitinib. To do so, we compared SPS levels in the 4 best responders (retrospectively defined as a complete response in skin and complete/near complete response internal organs) to the other 5 patients (Fig. 9e). We found that the best responders group tended to have lower baseline levels of SPS and tofacitinib resulted in relative normalization of SPS to levels comparable to healthy controls. In contrast, the other patients tended to have higher baseline SPS activity (or were heavily immunosuppressed when the baseline samples were collected, e.g. Pt4 and Pt5) with SPS levels tending to remain elevated above those seen in healthy controls even with tofacitinib (Fig. 9d).

**Evidence for dose-dependency of the response.** Given that relatively higher baseline SPS levels in plasma and persistent IFN-γ activity in skin were associated with partial improvement on tofacitinib, we hypothesized that in some patients, a higher dose of tofacitinib might be required. In order to explore this further, a single patient with partial improvement in cutaneous, pulmonary, and extra-thoracic lymph node sarcoidosis after 6 months of tofacitinib 5 mg twice daily during the trial (Fig. 9f, g) was subsequently treated with a higher dose of tofacitinib (10 mg in the morning and 5 mg at night) obtained through insurance after the trial. After an additional 6 months of treatment at the higher dose, there was complete clearance of his skin and after 8 months there was resolution of pulmonary and diffuse lymph node FDG-avidity (Fig. 9f, g). A relative normalization of the SPS signature was also observed on the higher dose (Fig. 9h). In an additional partial responder, we have since archived a complete cutaneous response by increasing tofacitinib to 10 mg twice daily. These data demonstrate dose responsiveness and suggest that a higher dose

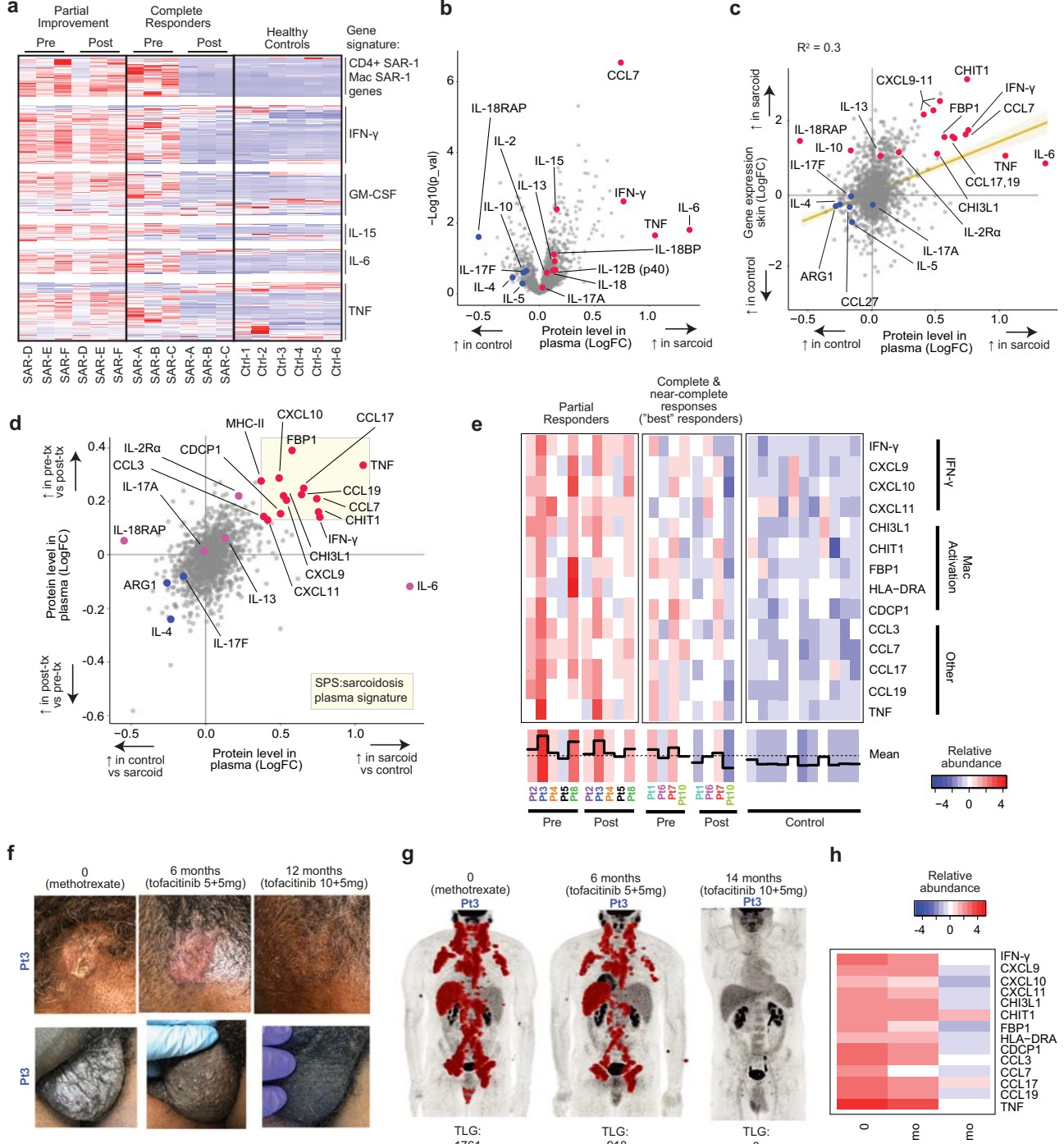

**Fig. 9 Tofacitinib reduces cytokine and chemokine levels and markers of macrophage activation in plasma. a** Heatmap showing expression of selected transcripts from bulk RNAseq of skin; sample labels are summarized in Supplementary Table 5. **b** Volcano plot showing relative abundance of proteins in plasma from sarcoidosis patients in this trial compared to healthy controls, $p$ values determined using t-tests, two-tailed. **c** Relative abundance of plasma proteins in sarcoidosis patients compared to controls (x-axis) compared to mRNA levels in skin of sarcoidosis patients (as determined by bulk-RNA-seq) compared to healthy controls (y-axis). Simple regression line with 95% confidence interval (shaded area) shown, goodness of fit calculated with R squared. **d** Relative abundance of proteins in plasma of sarcoidosis patients compared to healthy controls (x-axis) plotted against relative abundance of proteins in plasma of sarcoidosis patients at baseline (pre-tx) compared to after 6 months of tofacitinib (post-tx) (y-axis). The most consistently and highly differentially abundant proteins were used to create the sarcoidosis plasma signature (SPS) shown in the shaded box. **e** Heatmap showing plasma levels of SPS proteins in trial participants (before and after tofacitinib) and healthy controls, grouped by response pattern. Best responders include patients with a complete response in the skin (CSAMI 0 on treatment) and a complete or near complete response in other organs (98% or greater reduction in TLG). Partial responders include all other patients. **f** Clinical photographs of scalp and scrotal (biopsy-proven) sarcoidosis. Shown at baseline, after 6 months of tofacitinib at 5 mg twice daily, and after 12 months of tofacitinib 10 mg in the morning and 5 mg at night. **g** PET scans of the patient in panel **f** before tofacitinib, after 6 months at 5 mg twice daily, and after 14 months at 10 mg in the morning and 5 mg at night, TLG: total lesion glycolysis. **h** SPS levels corresponding to timepoints in (panel **f**).

of tofacitinib might be required in patients with more significant baseline disease activity.

## Discussion

In this open-label trial of tofacitinib, we demonstrate efficacy in 10 sarcoidosis patients with cutaneous involvement. In all 10 patients, disease control with a tofacitinib-based regimen was superior to the patients preceding immunotherapeutic regimen, particularly for skin involvement. Four of five patients entering the study taking prednisone were able to discontinue or significantly reduce the dose.

Our mechanistic evaluation suggested that IFN-γ is a key driver of sarcoidosis and is a critical cytokine targeted by tofacitinib with effective treatment. This observation is consistent with prior work showing that IFN-γ is elevated in circulation and in tissues of sarcoidosis patients and correlates with disease activity[37–40], and makes teleological sense given the fundamental role of IFN-γ in classical macrophage activation, granuloma formation, and protection against *Mycobacterium tuberculosis*[41,42]. Tofacitinib appears to provide an effective means of suppressing IFN-γ, which signals via JAK1/2, in sarcoidosis.

We also found that the activity of other cytokines including GM-CSF (JAK2), IL-15 (JAK1/3), IL-6 (JAK1/2), IL-12 (JAK2/TYK2) and TNF (JAK-independent) are also evident in sarcoidosis. GM-CSF has been shown to promote the differentiation of monocytes into inflammatory macrophages in autoimmunity[43], IL-15 can re-enforce CD4+ T cell effector responses[44], and IL-6 is an additional proinflammatory cytokine implicated as a potential treatment in sarcoidosis[45]. IL-12 has previously been implicated in sarcoidosis, however, interestingly, a clinical trial using ustekinumab, an inhibitor of p40 (both IL-12 and IL-23) was not effective in pulmonary sarcoidosis[8]. This suggests that isolated inhibition of IL-12 in sarcoidosis may be insufficient. Although we do not exclude a role for TNF in sarcoidosis, we do implicate these additional cytokines that may need to be inhibited for maximal response to therapy. A potential advantage of JAK inhibition (compared to TNF inhibition) is the simultaneous, direct inhibition of multiple cytokines. Furthermore, we observe a reduction in TNF production and activity with tofacitinib, suggesting that TNF production may occur, at least in part, downstream of these JAK-STAT dependent cytokines.

While all patients improved, some patients still had disease activity on tofacitinib. This was associated with incomplete suppression of IFN-γ, suggesting that higher doses might be required in some patients. Indeed, in a single patient, and later an additional patient, after the trial period, a higher dose of tofacitinib led to remission of disease. This is consistent with a prior case report, showing a better response of widespread sarcoidosis to tofacitinib 10 mg twice daily than to 5 mg twice daily[9]. Moving forward, improved suppression of IFN-γ activity could be achieved by either an increased dose of tofacitinib in some patients, or potentially, by evaluation of more targeted JAK inhibitors, such as a JAK1- or JAK1/2-specific inhibitors.

Limitations of the study include the small sample size and focus on cutaneous disease. Also, all patients had cutaneous involvement and thus may not be fully representative of sarcoidosis as a whole, where cutaneous involvement is present in about 1/3 of patients[46]. These data are promising and support further evaluation of JAK inhibition in the treatment of sarcoidosis.

## Methods

**Population and study design**. This is an open-label trial conducted at Yale University. Potentially eligible patients with sarcoidosis were identified in Yale Medicine clinics (dermatology, pulmonology, and cardiology). Inclusion criteria included: 1) a diagnosis of cutaneous sarcoidosis with supportive skin biopsy, 2) Cutaneous Sarcoidosis Activity and Morphology Instrument (CSAMI)[47] activity

score of 10 or greater, and 3) taking a stable dose of oral immunomodulatory regimen for at least 3 months, with no plans to otherwise change the regimen over the subsequent 6 months. CSAMI is a validated cutaneous sarcoidosis scoring tool that incorporates the degree of inflammation, induration, and surface change in lesions as well as the amount of skin affected into the score. In general, patients with a score of ≥10 would be considered candidates for systemic therapy based on the degree of their cutaneous involvement. Patients less than 18 years of age, with a history of chronic, untreated infections, and malignancy were excluded. Nine patients also had internal organ involvement, including pulmonary involvement (3 patients had a history of Scadding stage 1 disease and 6 patients a history of Scadding stage 2 disease) (Table 1 and Supplementary Table 1), but presence or absence of internal organ involvement was not used as a criterion to determine eligibility. All participants, including the healthy controls, provided written informed consent, including consent to publish clinical images. Participants were compensated 50 USD for each clinic visit and 250 USD for each PET scan. The data was collected at the Yale Center for Clinical Investigation at Yale University School of Medicine in New Haven, Connecticut, United States. The first patient was enrolled on April 11, 2019 and the last patient was enrolled on May 8, 2020.

**Study treatment and endpoints**. There was no washout period and baseline data was obtained while the patients were taking their preceding immunosuppressive regimen. Tofacitinib 5 mg twice daily was initiated, and patients were evaluated after 1, 3, and 6 months of treatment. The primary endpoint was change in the CSAMI activity score after 6 months. Change in internal organ disease activity was determined by whole body PET – computed tomography (CT)[9]. Total lesion glycolysis (TLG), which is the product of the metabolic lesion volume and mean activity, was used to quantify internal organ disease activity. TLG was determined before and after 6 months of treatment with tofacitinib and is discussed in greater detail below. Change in skin-related quality of life (Skindex-16)[48] was also assessed. Molecular parameters of disease were assessed at baseline and after 6 months of treatment and are described in greater detail below.

**Management of other immunosuppression**. Most patients were on therapy for their sarcoidosis at the outset of the study (Fig. 1a). Patients had the option to continue, change the dose, or discontinue other immunosuppressive medications at the outset, or during the study. One patient was taking hydroxychloroquine and it was discontinued upon initiation of tofacitinib. Four patients were taking methotrexate at the outset, all elected to discontinue this medication upon initiation of the tofacitinib. Five patients were taking prednisone. Throughout the 6-month study period, patients were permitted to taper and/or discontinue the prednisone based on their symptoms, physical examination findings, and the investigators' discretion. In one patient, prednisone was increased during the study period related to increased pain associated with pre-existing Achilles tendinopathy, which had been responsive to prednisone.

**Study oversight**. The study and all uses of human material were approved by the Yale Institutional Review Board and performed in accordance with the protocol and with the principles of the Declaration of Helsinki and local regulations. All authors had access to the primary data and approved the decision to submit the manuscript. The trial was registered with clinicaltrials.gov (NCT03910543).

**Dermatologic evaluation and specimens**. The clinical extent of cutaneous sarcoidosis was quantified using the validated Cutaneous Sarcoidosis Activity and Morphology Instrument (CSAMI)[47], which quantifies disease activity (representing active inflammation) and tissue damage (representing tissue damage/destruction) separately. Only the activity portion of the CSAMI instrument was utilized for this study. Two optional 4 mm punch biopsies of lesional skin were obtained from each patient, prior to initiation of tofacitinib (while taking the preceding immunomodulatory treatment regimen) and again after 6 months of tofacitinib. In patients where disease activity remained on tofacitinib, a lesion that still appeared active was biopsied. In patients with an apparent complete clinical response to tofacitinib, a representative, previously active area of skin was biopsied. In patients with primarily facial involvement, the research biopsies were generally deferred.

A portion of the biopsy tissue from each patient was placed in RNAlater reagent (Qiagen) and snap frozen for downstream analyses (see below) and another portion was fixed in 10% buffered formalin and embedded in paraffin for histologic analysis (see below). In 3 patients, a third biopsy was obtained from lesional skin prior to initiation of tofacitinib and was used for single cell RNA sequencing (see below). Skindex-16 is a validated, skin-related quality of life tool[48]. Skindex-16 was administered at baseline and after 6 months of tofacitinib therapy.

In the skin, complete response was defined as reduction of the CSAMI activity score to zero after 6 months of therapy. Partial response in the skin was defined as any other reduction in CSAMI to a value other than zero after 6 months of therapy.

**FDG-PET/CT**. FDG-PET imaging was performed in the morning following a one-day high fat/low carbohydrate diet followed by an overnight fast of greater than 12 h as previously described[49]. Dietary preparation instructions were provided in writing to the patient, along with telephone contact by clinical staff 48–72 h prior to the study.

Both the cardiac Rubidium-82 (Rb-82) and whole-body FDG-PET imaging were performed on a GE Discovery D690 PET/CT scanner. Low-dose CT images (120 kV, 50 mA for BMI < 39, modulated at 50–150 mA for BMI > 39) for the purposes of attenuation correction were performed before Rb82 and FDG imaging sequences.

Resting ECG-gated dynamic Rb-82 PET imaging were performed using 20–35 millicurie (mCi) of Rb82 consistent with established clinical guidelines and our previous publications[50]. Following Rb-82 imaging, the patient was injected with FDG (8–10 mCi), followed by a 60 min waiting period during which the patient relaxed or read in a quiet room. Whole-body FDG images were acquired for 2.5 min per bed position from cranial apex to knee, and 2 min per bed position from knees to toes. A separate, 8 min single bed position cardiac acquisition was then performed (cardiac protocol described further below).

Resting Rb-82 perfusion imaging was processed and interpreted visually with 4DM (Invia). Quantitative and visual analysis of cardiac FDG uptake was performed on the AW software platform (GE) on the dedicated 3D cardiac acquisition window (single 47 slice bed position) and reported as previously described[51,52]. (Quantitative and visual analysis of extra-cardiac FDG update is described below.) When present, the relationship between the location of inflammation and perfusion defects was characterized. The maximum standardized uptake value (SUV) in the heart was used to represent the peak level of inflammation. The volume of inflamed myocardium, cardiac metabolic volume (CMV), was identified as myocardium exceeding a SUV threshold that was derived for each patient by multiplying the left ventricular (LV) blood pool (background) activity by 1.5 as previously described[49]. Cardiac metabolic activity (CMA) was defined as the CMV multiplied by the average SUV of that volume.

At the time of six-month follow up imaging, patients were provided with their written diet log from the initial scan and instructed to replicate the same high-fat/low carbohydrate diet and fast duration as closely as was feasible. Follow up scans were performed using the same protocol and PET scanner.

The sarcoidosis metabolic lesion burden at baseline and post-treatment follow-up was determined using the whole-body PET segmentation tool LesionID® v 7.0.4 (MIM Software Inc, Cleveland, Ohio, U.S.A.). This tool allows semi-automatic determination of maximal and mean standardized uptake values (SUV) ($SUV_{max}$ and $SUV_{mean}$, respectively), metabolic tumor volume (MTV, $cm^3$), and total lesion glycolysis (TLG, product of $SUV_{mean}$ and MTV) of individual sarcoidosis lesions, and subsequent automatic calculation of MTV and TLG for the whole body. In the absence of a dedicated inflammatory lesion lexicon, the oncology lexicon was used to define metabolic parameters for sarcoidosis burden assessment.

The initial automatic segmentation (lesion contouring) performed by the software was based on a predefined threshold which was defined by placing a 3 cm diameter sphere in the right lobe of the liver. The threshold value was determined using 1.5 * liver $SUV_{mean}$ + 2 * liver standard deviation, as defined by PET Response Criteria in Solid Tumors (PERCIST)[53]. All voxels above this value were segmented and volumetrically separate regions were made into single volumes of interest (VOIs). The automatic segmentation was subsequently corrected by consensus of 3 readers, a dual-boarded radiologist and nuclear medicine physician (15 years of experience) (D.P.), a board-certified cardiologist (B.D.Y.), and a board-certified dermatologist (W.D.). During the correction process, the readers rejected false-positive lesions (mostly attributed to physiological uptake or pathologic uptake deemed sarcoidosis-unrelated) and made any necessary additional edits (modifying segmentation to avoid physiological uptake). All regions were combined into a total lesion burden VOI that encompassed all approved lesions for that timepoint.

All lesions were tracked across time at the follow-up study. On follow-up scan, we placed a liver reference region in the right lobe of the liver to calculate the threshold value specific to this timepoint. Tracked lesions were transferred from the baseline PET/CT to the follow-up PET/CT via a rigid fusion and redefined automatically using PET Edge+®, a hybrid intensity and gradient-based tool[54]. We adjusted the tracked lesion VOIs as described above, if necessary. Next, similarly to the baseline timepoint, the rest of the image volume, not including the tracked lesions, was segmented with the PERCIST value specific to the follow-up timepoint. We followed the same steps as described above on the baseline timepoint to approve and finalize a total lesion burden VOI for the follow-up timepoint.

The software then calculated all metabolic parameters from the total lesion burden VOI based on final lesion contours created by semi-automatic segmentation and individually tracked lesions were compared across both timepoints. Baseline and follow-up studies were processed simultaneously for immediate automatic determination of treatment-related changes in the individual patients.

For internal organ involvement, complete/near complete response was defined retrospectively as reduction of TLG by 98% or greater after 6 months of therapy. Partial response was defined as any reduction in the TLG less than 98% after 6 months of therapy.

**Assessment of myocardial involvement in patient 1.** In one patient (Pt1), the only disease activity that was evident at the beginning of the study was myocardial activity visualized on the dedicated cardiac scan. This myocardial activity was not above threshold in the whole-body PET-CT scan (and so TLG is shown as 0 in Fig. 1c). However, bonafide myocardial disease activity was evident on the

dedicated cardiac scan, as interpreted by a board-certified cardiologist and director of the Nuclear Cardiology lab at Yale (E.J.M.). This activity was quantified separately, as described in the previous section, and is shown in Fig. 2e. Although all patients received dedicated cardiac scans before and after 6 months of tofacitinib, only this one patient had a positive test. In all other patients, the dedicated cardiac studies were negative, both before and after 6 months of treatment.

**Histology, immunohistochemistry, and RNA in situ hybridization.** Histology and immunohistochemistry (IHC) were performed on formalin-fixed, paraffin-embedded sections using standard methods. For IHC, antigen retrieval was performed using citrate buffer (pH 6.0) (Life Technologies). The CD68 antibody was obtained from Dako (PG-M1) and used at a dilution of 1:100. Primary antibody binding was detected and visualized using ImmPRESS peroxidase reagent kit (Vector) and diaminobenzidine (DAB) substrate (Vector).

RNA in situ hybridization was performed using the RNAscope platform (Bio-techne) as previously described in detail[35]. Probes for *IFNG* (Cat #310501), *IL12B* (Cat #402071), *IL13* (Cat #586241), and *IL17A* (Cat # 310931) are pre-diluted, were purchased directly from the manufacturer (Bio-techne), and used according to the manufacturer's specifications. Samples analyzed using this approach are summarized in Supplementary Table 6. Quantification was performed in a blinded fashion by a board-certified dermatopathologist (W.D.) using a slight modification of the guidelines established by the manufacturer to allow cell-by-cell scoring and described as follows. A score was assigned for each cell in the tissue and ranged from 0 to 4. A cell received a score of 0 if there was no staining or < 1 dot / cell. A cell received a score of 1 if there were 1–3 dots/cell. A cell received a score of 2 if there were 4–9 dots per cell. A cell received a score of 3 if there were 10–15 dots per well. And a cell received a score of 4 if there were >15 dots per cell. Positive cells having scores of 2, 3, or 4 and were included in the analysis. Cells with scores of 0–1 were not quantitated and in our experience represent background staining[35]. The total number of positive cells per $mm^2$ of tissue was calculated for each case.

**Bulk RNA extraction, sequencing, and analysis.** Skin biopsy tissue stored in RNAlater reagent (Qiagen) was thawed on ice. The tissue was homogenized using a rotor-homogenizer (PowerGen 125, Fisher Scientific). Total RNA was extracted using the RNeasy Fibrous Tissue Mini Kit (Qiagen) according to the manufacturer's instructions. Samples analyzed using this approach are summarized in (Supplementary Table 5). RNA was submitted to the Yale Stem Cell Center and complementary DNA (cDNA) libraries were generated and 100-basepair paired-end sequencing was performed on an Illumina HiSeq 4000 at Yale Stem Cell Center.

Bulk RNA-seq data was initially processed using Partek Flow software (Version 9.0, build 9.0.20.0510). Burroughs-Wheeler Aligner (BWA, Version 0.7.15) was used to align the reads against hg38 (hg38_refseq_16_02_02 was used for quantification). Additional, previously published healthy control skin RNAseq samples run on the same platform were also included (GEO Accession number GSE122592)[55]. Reads were normalized by counts per million (CPM). Further processing to identify differentially expressed genes between control, pre-treatment, and during treatment groups was carried out using DESeq2. Target gene transcript lists were downloaded from Gene Set Enrichment Analysis Molecular Signatures Database or Ingenuity Pathway Analysis[56].

**Single cell RNA sequencing (scRNA-seq).** 4 mm punch biopsies of skin were obtained from 3 patients with sarcoidosis and 3 healthy volunteers (Supplementary Table 3). Biopsies were immediately placed in to Dispase II (10 mg/mL) (Sigma) in RPMI with 2% FBS for 45 min at 37 °C with shaking at 125 RPM. The dispase solution was placed on ice. The tissue was removed from the dispase solution, minced in a sterile plate with a scalpel, and then subsequently incubated in Liberase TH (0.5 mg/mL) (Sigma) in RPMI with 2% FBS for 45 min at 37 °C with shaking at 125 RPM. After incubation in the liberase and the dispase solutions were combined (to recover any additional cells from the dispase) and the mixture was triturated and filtered through a 100 μM sterile filter. Cells were washed with RPMI + 2% FBS and erythrocytes were lysed with ACK Lysing Buffer (Lonza) and washed again in RPMI + 2% FBS. Cells were stained with LIVE/DEAD Red viability dye (ThermoFisher L23102) at a 1:1000 dilution in RPMI + 0.1% FBS. Viable single cells were isolated using a BD FACS Aria cell sorter and sorted into cold RPMI + 2% FBS. The gating strategy is shown in Supplementary Fig. 2. Up to 10,000 cells (depending on the yield) were loaded onto a 10X Chromium Controller. Single cell libraries were prepared according to the manufacturer's instructions by the Yale Center for Genome Analysis (YCGA). Sequencing was performed with one sample/library per lane on an Illumina Hiseq 4000 by YCGA.

Cellranger software (v3.1.0) was used to process the data. Cellranger mkfastq was used for processing raw files into fastq files, and cellranger count was used to align reads to hg38, filter the reads and generate a cell-by-gene matrix for each library. Libraries were aggregated using cellranger aggr without normalization to generate a single cell-by-gene matrix. The Seurat R package (v3.2.0) was used for analyzing skin samples. Droplets with ≤ 100 expressed genes were removed from the matrix. The NormalizeData command with a scaling factor of 10,000 was used to normalize counts. ScaleData was used to regress the data against the number of transcripts and center gene expression values. Principle component analysis (PCA)

was performed using RunPCA. Cells were clustered using FindNeighbors and FindClusters. Clusters consisting of cells with low/null expression of *GAPDH* (non-cells) were removed from further analysis using the SubsetData command, resulting in 24,034 cells for analysis. For lung BAL samples, we analyzed scRNA-seq data from Liao et al. (n = 4 sarcoidosis BAL)[29] and Mould et al. (n = 10 control BAL)[30]. To minimize potential batch effects, we normalized reads with SCTransform and integrated the data using the reciprocal PCA method with Seurat (v4.0.4) for analysis of the pulmonary sarcoidosis samples. The resulting data was visualized by performing Uniform Manifold Approximation and Projection (UMAP) dimensional reduction.

Cell-type assignments for each cluster were determined using canonical markers. For immune cell subsets, cell type assignments for each cluster were verified by comparing with ImmGen datasets[57]. T cell, myeloid cell, and fibroblast clusters were subsetted, and singlets were re-analyzed separately using the approach described above. Stacked histogram plots were generated in R (Version 3.6.0) using ggplot2 (v3.3.3), ggrepl (v0.9.1) and dplyr (v1.0.3) were used to generate dot plots.

**Pathway analysis, cellphone DB, and data visualization**. Ingenuity Pathway Analysis (IPA, Ingenuity Systems Qiagen, Content version: 51963813) was used to perform core analyses on both bulk RNA-seq data and scRNA-seq data. Canonical pathways, upstream regulator and causal network analyses were utilized to compare differentially expressed gene lists generated using Seurat (scRNA-seq) or Partek Flow Version 9.0 (bulk RNAseq). ggplot2 v3.3.3, ggrepl (v0.9.1) and dplyr (v1.0.3) were used to visualize the results by plotting Z-score versus p-value. Selected genes or pathways were labeled at the investigators' discretion. Heatmaps were generated using gplots (v3.0.4) heatmap.2 function. Raw values were scaled using the scale function. Manhattan clustering was utilized. For Cellphone DB (version 2.1.4), normalized counts and meta data were exported from Seurat and imported into CellphoneDB. The statistical_analysis command was used. Dotplots were generated using the dot_plot command.

**Multiplex proteomic analysis of plasma**. Blood samples were collected from participants at baseline and again after 6 months of therapy. Samples were compared to 11 healthy, roughly age matched controls (Supplementary Table 7). Blood was collected in EDTA-coated tubes and centrifuged; the buffy coat was removed and stored separately from the plasma, which was aliquoted and stored at −80 °C. Plasma was analyzed using the Olink Explore 1536 panel. This is a multiplexed panel that quantifies 1536 protein analytes simultaneously using a proximity extension approach. Detection of individual proteins requires binding by two matched, oligo-barcoded antibody pairs for each target. When both antibodies are bound, the oligo tags come in close proximity to each other and hybridize forming a unique template for DNA polymerase-dependent extension. After PCR amplification, next generation sequencing is performed and processed by Olink (Boston, Massachusetts). Log-fold changes and p values were calculated for pre- vs post-treatment and pre-treatment vs healthy. The data was then imported into R (Version 3.6.0) to generate heatmaps and volcano plots using ggplot2 (v3.3.3), ggrepl (v0.9.1), gplots (v3.1.1) and dplyr (v1.0.3).

**Analysis of bulk gene expression in BAL samples**. We accessed and downloaded previously published bulk gene expression data from sarcoidosis BAL samples in the GRADS dataset[32] and from healthy control BAL samples dataset[33]. There were 76 patients in the GRADS dataset with untreated sarcoidosis available for analysis, including 24 untreated Scadding stage 1, 40 untreated stage 2/3, and 12 untreated stage 4 patients. We downloaded and analyzed the FPKM data contained in the file 'GSE109516_SARCOIDOSIS_fpkmdata_BAL_clean_withannot_ gradsids_added.tsv' for these sarcoidosis patients. A total of 6 healthy control patients had data available for analysis from the GSE136587 dataset. FPKM data was not provided, so we uploaded the raw data into Partek Flow version 9 using the BioProject ID PRJNA562768. We performed quality control by trimming low quality bases from the 3′ ends and removing low quality reads; this was done using the 'trim bases' tool with the default settings. Next, we mapped reads to the hg38 reference genome using STAR and quantified the aligned reads to an annotation model. Gene counts were then normalized using Cufflinks to calculate FPKMs. FPKM data for sarcoidosis and control samples was imported into Python (Version 3.8.8) and gene expression levels were visualized and compared using the boxplot and swarmplot functions from Seaborn (v0.11.1).

**Analysis of gene expression data from FACS purified myeloid populations**. We accessed and downloaded previously published data consisting of FACS sorted myeloid populations from BAL of patients with sarcoidosis and healthy controls (GSE174659)[34]. Data was available for 9 healthy controls (HB02, HB16, HB24, HB25, HB26, HB27, HB28, HB29, HB55) and 8 patients with sarcoidosis (SB36, SB37, SB39, SB51, SB53, SB63, SB73, SB89). We downloaded and analyzed the batch-corrected counts data contained in the file 'GSE174659_Counts_batch_removed.csv'. Data was imported into Python and gene expression levels were visualized using the boxplot and swarmplot functions from Seaborn (v0.11.1).

Differences in gene expression between healthy controls and patients with sarcoidosis were assessed for significance using the unpaired t-test in Graphpad Prism.

The identity of FACS sorted populations, as described in the initial publications, were confirmed using canonical markers in our re-analysis of the data. Data on each population was not available for each sample (all of the data that was available was analyzed).

**Statistical analysis**. Statistical analysis was performed using GraphPad Prism 9. Simple linear regression analysis and p value determination was performed. The 95% confidence interval bands were plotted in Prism. Line graphs and histograms were also plotted in Prism. P values were calculated using an unpaired t test.

**Reporting summary**. Further information on research design is available in the Nature Research Reporting Summary linked to this article.

## Data availability
The data generated in this study have been deposited into Gene Expression Omnibus under accession numbers as summarized in Supplementary Table 8. The clinical data are protected and are not available due to data privacy laws. Source data are provided with this paper.

## Code availability
All code used to generate and analyze the data in this study utilize standard commands with publicly available platforms; no custom code was utilized.

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

## Acknowledgements

The study was funded in part through an investigator-initiated research grant from Pfizer to Yale (W.D.). The authors would like to thank G. Wang and the Yale Center for Genome Analysis as well as M. Zhong and the Yale Stem Cell Center sequencing core for assistance with scRNA-seq and bulk RNA-seq experiments, respectively. We also acknowledge S. Liao, L. Maier, and I. Yang and collaborators for sharing published pulmonary scRNAseq data with us. We thank B. Olamiju as well as the Foundation for Sarcoidosis Research for their assistance with patient recruitment. W.D. was supported by Career Development Awards from the Dermatology Foundation and from NIAMS (K08AI159229-01) and a grant from the Robert E. Leet and Clara Guthrie Patterson Trust. I.D.O. was supported by a Career Development Award from the Dermatology Foundation. C.R. is supported by research funding from the Chest Foundation and NIH/NHLBI (1K08HL151970-01). D.J.K was a Paul and Daisy Soros Fellow and was supported in part by a grant from the National Cancer Institute and National Institutes of Health (NIH) (F30CA236466) and an MSTP training grant (T32GM007205). B.K. is supported by the Ranjini and Ajay Poddar Fund for Dermatologic Diseases Research.

## Author contributions

W.D. and B.K. designed the study. W.D. performed the clinical trial. A.W., D.K, K.S., M.J.M, J.D, I.D.O, and M.K.M. performed experiments. A.W., D.K, M.J.M., A.C., R.A., analyzed the data. C.R., R.H., M.G, provided key patient data and patient samples. B.D.Y, R.F.C, D.P, and E.J.M. designed, collected, interpreted, and analyzed the imaging studies. M.W.B. and R.A. supervised portions of the research. W.D. supervised the majority of the research. W.D. wrote the paper. W.D. and B.K. conceived of the project.

## Competing interests

W.D. has research funding from Pfizer and Advanced Cell Diagnostics/Bio-techne, serves as a consultant for Eli Lilly, Pfizer, Incyte, and Twi Biotechnology, and receives licensing fees from EMD/Millipore/Sigma. B.D.Y receives research funding from Pfizer. E.J.M. receives research funding from Alnylam, Pfizer, and Eidos and serves as a consultant for Alnylam, Pfizer, and Eidos. D.P. receives consulting fees from Telix Pharmaceuticals and Cohere Health. M.B. is a consultant for Eli Lilly and receives licensing fees from EMD/Millipore//Sigma. R.A.F. is a consultant for Glaxo Smith Kline and Zai labs. B.K. is a

consultant to and/or has served on advisory boards for Aclaris Therapeutics, Arena Pharmaceuticals, Bristol-Meyers Squibb, Concert Pharmaceuticals Inc, Dermavant Sciences, Eli Lilly and Company, Pfizer, and VielaBio; he is on speaker's bureau for Pfizer, Regeneron and Sanofi Genzyme. A.W., D.J.K., K.S., M.J.M., J.D., A.C., R.A., C.R., M.K.M., I.D.O., R.F.C., R.H. and M.G. have no disclosures.
