## [Peer Review File · Nature Communications]

REVIEWER COMMENTS

Reviewer #1 (Remarks to the Author):

In this report by Damsky W et al, the authors seek to examine the response to Tofacitinib, a JAK1,2,3 inhibitor for 10 patients with sarcoidosis, in an open-label interventional study, and the key pathways targeted by this drug in some of these patients.

The main strengths of this paper are - (i) the combination of molecular findings and support for clinical observations, using advance and comprehensive screening methods (scRNAseq and proteomics) (ii) use of disease activity methods (CSAMI and TLG) (iii) well written, analysed and presented. The scRNA seq findings are extremely interesting and the single cell transcriptomic part represents a substantial piece of work for sarcoidosis; though not necessarily for the aim of this study (see later).

Main issues are the lack of clinical context for the disease, the limitations are not clearly discussed and specified, the study findings are not quite novel given publications by the same authors and another group(ref 14-18) on the same type of patients. Overall, the clinical part is relatively weak and as this is the input to the stronger molecular work, the interpretation is needs to be framed within the limitation of the clinical 'substrate'.

Major comments (in order of appearance in paper, and not necessarily in order of significance). If some of the queries here are addressed in Suppl, then it needs to come into text as the suppl data are too hefty for clear interpretation of findings in main text.

1. This is a very skin-biased view of sarcoidosis, and patients have predominantly skin issues. It is important to put this in context of the disease which is a very heterogenous disease, with many different clinical manifestations, of which the skin is arguably the least important (cosmesis aside). The life threatening and most important component of sarcoidosis are more often the progressive fibrotic lung disease and the severe cardiac or neuro disease. For the benefit of the readership, this should be clearly indicated - that you are looking at predominantly cutaneous sarcoidosis. e.g. line 92 - ' .. long standing recalcitrant sarcoidosis with cutaneous involvement...' should be replaced with 'long standing cutaneous sarcoidosis'.

2. Overall patient profiles are rather messy. Please clarify these queries:

- line 92 'recalcitrant' - unclear how this can be the case when patient 9 has had condition for only one year; and patient 1 was on no treatment at baseline? Please define 'recalcitrant' - is this presence of skin lesions despite how many years of treatment. What does 'duration' in Table 1 refer to - is it duration of treatment or disease?
- what is skin pattern of 'dermal' and 'subcutaneous' in table 1 - please clarify in table or text. Lupus pernio and massive multiple plaques are much more severe form of disease compared to occasional skin lesion as perhaps in Patient 7. Can this be expanded briefly in Table 1 given the small number of patients, and the unusual range of treatment at baseline (from no treatment in patient 1 and 7 ; to pred and MTX in patient 6). Duration of treatment also important to give a view of how resistant the disease has been to treatment.

3. Briefly specify what CSAMI and TGL measure please in main text . CSAMI range at baseline is large 20-57 - please clarify what this means.

4. Ethnicity of patients (more than skin phototype) required as the type of sarcoidosis disease in Afro-Caribbean, Caucasians etc are different

5. Line 139 - how was lesions selected for skin biopsy, please confirm biopsy post treatment is from the same site - if not, please state why not and where skin biopsies were obtained

6. The term 'responders' 'partial' and 'complete' responders need clarification and definition. This needs to be clearly stated from the beginning. line 142; 242 (what is 'partial' responder); 266 (what is 'best' responders - pt 2,3,4,5,8 were classed partial responders and pt 1,6,7,10 as complete/near complete responders (Fig4D) on what basis please? Why is pt 3 classified as complete responders in Fig 3J)?
6. Line 146 - 8 patients had pulmonary involvement - Table 1 states only 4 had lung involvement - did you count thoracic lymphadenopathy as lung involvement? If so, please state as this is debatable; most clinicians would view pulmonary involvement as presence of interstitial lung disease.
7. table 1 please state what is n/a - not assessed? not available?
8. Figure 1E - unclear how dactylitis is visible on picture (!!) and not mentioned in Table 1. the only change visible in Fig 1E appears to be in toe nails?
8. Figure 1 G - pls specify where the biopsy is from (cf point 5 above)
9. Figure 2A. Where is Pt 1 on graph pls?
10. Figure 3A and scRNAseq studies:
 - this is the main strength of the paper but limitations need to be stated. Only 3 patients in a heterogenous disease and patient numbers (1,2,5) need to be stated upfront in the text (not suppl data) and the fact that one is on no treatment and the other two on MTX and Pred (for ?years) need to be specified as this will likely impact on gene expression profile.
 - Figure 3 B and E - confusing. What is "SAR other" please? It seems also that cluster 2,7 and 12 are over represented differentially in all three patients? Fig 3C - how were these transcripts selected? pls specify. Suggest move Figure S3 to main figure to address this.
11. Fig 4C - left lower quadrant probably just as important as right upper quadrant.

Reviewer #2 (Remarks to the Author):

The authors have submitted a report on an open label study of Tofacitinib in sarcoidosis, accompanied by supporting laboratory investigations to provide mechanistic explanation for the drug.

The clinical study is well described. The authors sought local approvals to undertake the work. There was sponsorship from the company that manufactures Tofacitinib.

The 10 patients are representative of a severe sarcoid cohort. Table 1 is important in understanding the disease patterns. One question - I was unsure on the difference between 'no' and 'n/a' in the Other column.

The supplementary data are excellent, and have answered all the questions I had regarding the population, outcomes and adverse event profiles.

The PET images and skin photographs provide useful context, and the images show very typical sarcoid disease manifestations.

The paired laboratory data follow a logical flow. I have looked through each figure, and the plots are easily interpretable, and all appear genuine.

The cytokine profiling is consistent with the authors' hypothesis regards the IFN suppressive effects of Tofacitinib.

Overall, the manuscript is a very compelling argument to undertake further study of this therapeutic strategy in sarcoid.

Reviewer #3 (Remarks to the Author):

This report describes an open label trial where 10 patients with sarcoid recalcitrant to other treatments were with 5 mg of tofacitinib twice a day, without stopping other treatments or washout. Response was assessed using a skin-focused clinical activity and morphology score at 6 months and also by TLG assessed by PET. Greater than 80% CSAMI improvement was observed, with 6 patients showing a complete response – this number was 3/10 for TLG:PET.

scRNAseq analyses of affected tissue before and after treatment pointed to inhibition of CD4+ T cell interferon production as a cause for improvement. Specifically, CD4+ T cell clusters were identified for 3 sarcoid patient biopsies not found in normal controls, and in one such cluster, IFNG was the most upregulated cytokine, compared to modest TNF and minimal IL-17 expression. IFNG bias was clear in the overall scRNAseq analysis for sarcoid samples and normalized with tofacitinib treatment. In the APC analysis, the most interesting data likely regarded a population of cDC1-like cells showing transcript expression of IL12 in the sarcoid samples.

A series of serum sarcoid inflammatory markers were also found in correlation with disease severity and response to jakinib, including CXCL9, CXCL10, and CXCL11.

It was challenging for this reviewer that the title and first half the abstract read like a clinical study, and that the second half of the abstract veers into molecular analysis. It is really only this second half that has the potential for novelty, but as described below, I have serious questions about the depth and rigor of this deeper molecular investigation.

I think the authors are aware that the real question is not whether these findings are clinically significant in a broad sense. (They are.) Rather, the question regarding whether this paper belongs at Nature Communications centers on whether something distinctly new is brought to the table. There has been a recent run of valuable studies on treatment of sarcoid using jakinibs including tofacitinib, none as prominent as one by these authors in NEJM.

In that NEJM article, not only was substantial improvement of sarcoidosis in response to tofacitinib noted, but the bulk RNAseq results showed “While the patient was taking tofacitinib, there was down-regulation of the messenger RNA (mRNA) in the JAK-STAT–dependent pathways (interferon- γ and interleukin-6) as well as of mRNA in pathways that are not directly regulated by JAK-STAT (TNF α and mTORC1).”

Regarding pre-and post-treatment, the NEJM paper notes: Gene-set enrichment analysis showed activation of interferon- γ and tumor necrosis factor (TNF) α signaling as well as interleukin-6–STAT3 and mammalian target of rapamycin complex 1 (mTORC1) signaling in the pretreatment skin-lesion samples; in contrast, in skin-lesion samples obtained during treatment and in normal skin samples, these pathways did not appear to be activated.”

The cytokine list is longer in this study, but the precision is not much greater.

The single-cell analysis here is simplistic, focusing on CD4 and CD8 categories. Problematically, there are virtually no CD8 cells in the controls and it may be that all control CD4 cells are clustered separately into a cell-type called “Ctrl”? It would be far more

reassuring to recognize different resident memory populations, levels of exhaustion in CD8 cells etc. I came away understanding for sure that IFNG and CSF2 are up in CD4 cells in sarcoid lesions and are normalized during treatment and correlate with response. But what does this bring to a casual or expert audience vs. the previously published papers? I think there is some novelty in this study, on a level well served in JID or JACI.

Reviewer #1 (Remarks to the Author):

In this report by Damsky W et al, the authors seek to examine the response to Tofacitinib, a JAK1,2,3 inhibitor for 10 patients with sarcoidosis, in an open-label interventional study, and the key pathways targeted by this drug in some of these patients.

The main strengths of this paper are - (i) the combination of molecular findings and support for clinical observations, using advance and comprehensive screening methods (scRNAseq and proteomics) (ii) use of disease activity methods (CSAMI and TLG) (iii) well written, analysed and presented. The scRNA seq findings are extremely interesting and the single cell transcriptomic part represents a substantial piece of work for sarcoidosis; though not necessarily for the aim of this study (see later).

Main issues are the lack of clinical context for the disease, the limitations are not clearly discussed and specified, the study findings are not quite novel given publications by the same authors and another group (ref 14-18) on the same type of patients. Overall, the clinical part is relatively weak and as this is the input to the stronger molecular work, the interpretation is needs to be framed within the limitation of the clinical 'substrate'.

Major comments (in order of appearance in paper, and not necessarily in order of significance). If some of the queries here are addressed in Suppl, then it needs to come into text as the suppl data are too hefty for clear interpretation of findings in main text.

1. This is a very skin-biased view of sarcoidosis, and patients have predominantly skin issues. It is important to put this in context of the disease which is a very heterogenous disease, with many different clinical manifestations, of which the skin is arguably the least important (cosmesis aside). The life threatening and most important component of sarcoidosis are more often the progressive fibrotic lung disease and the severe cardiac or neuro disease. For the benefit of the readership, this should be clearly indicated - that you are looking at predominantly cutaneous sarcoidosis. e.g. line 92 - ' .. long standing recalcitrant sarcoidosis with cutaneous involvement...' should be replaced with 'long standing cutaneous sarcoidosis'.

We appreciate the feedback from Reviewer #1 and the comment about the focus on the cutaneous involvement. We have made the following changes to the text to address this concern:

- 1. We have updated wording in specific areas as suggested.*
- 2. We have added text to the discussion stating that this cohort (where all patients had skin involvement) might not be fully representative of all patients with sarcoidosis [Roughly 1/3 of patients with sarcoidosis have skin involvement].*
- 3. We have added text to the discussion stating that limitations of the study include the small number of patients and the focus on cutaneous disease. However, we do note that 9/10 patients had internal organ involvement – including thoracic lymph nodes, extra-thoracic lymph nodes, lung parenchyma, spleen, and myocardial involvement – involvement which was followed in 8/9 patients.*
- 4. We have performed a series of new molecular analyses to broaden the scope of the immunologic analysis beyond the skin and consider the clinical heterogeneity in this disease. This has led to devoting 3 main figures (as opposed to only 2) to the molecular analyses in the revised version.*

- a. We reanalyzed publicly available transcriptomic data from bronchoalveolar lavage (BAL) samples from 76 pulmonary sarcoidosis patients from the GRADS cohort (Vikmirovic et al. *Eur Resp J* 2021, PMID: 34083402). This data set encompasses untreated patients with diverse pulmonary phenotypes including Scadding stage 1 (n=24), stage 2/3 (n=40), and stage 4 (n=12). These data were compared to data from 6 healthy controls (Camiolo et al. *Cell Rep* 2021, PMID: 33852838). These analyses help to validate our molecular findings outside the skin, in a larger number of patients, in a clinically diverse cohort. See **Figure 3b and Supplementary Fig. 9**.
- b. We have expanded discussion of our analysis of a separate publicly available transcriptomics data set from cutaneous sarcoidosis (to further address disease heterogeneity in the skin). This data set includes gene expression data from skin biopsies of 15 patients with cutaneous sarcoidosis versus controls. These analyses help to corroborate our molecular findings in a larger number of patients with cutaneous sarcoidosis. See **Figure 3a**.
- c. We reanalyzed publicly available scRNA-seq data from BAL from 4 patients with pulmonary sarcoidosis (Liao et al. *Eur Resp J* 2021, PMID: 33602861) and 10 healthy control patients (Mould et al. *Am J Respir Crit Care Med* 2021, PMID: 33079572). These analyses help to corroborate our scRNA-seq findings outside the skin; in particular we find CD4+ T cells with a very similar phenotype to the CD4+ T cells found in skin. See **Supplementary Fig. 8**.
- d. We reanalyzed publicly available gene expression data from FACS purified myeloid cell populations from sarcoidosis patients (n=8) and healthy control patients (n=9) (Lepzien et al. *Eur Resp J* 2021, PMID: 33446605). These analyses also help to corroborate our molecular findings, including findings from our cutaneous scRNA-seq data, outside the skin and in a larger number of patients. See **Figure 3c and Supplementary Fig. 10**.
- e. We performed cytokine staining on additional cases of cutaneous (n=10) and pulmonary (n=10) sarcoidosis relative to controls in order to validate key cytokine signals in additional patients' samples. See **Figure 3d and 3e**.

**Importantly, in all analyses of publicly available data sets, we reach conclusions that are distinct from the original publication; this generally occurred by comparing data sets in sarcoidosis patients to separate, comparable control data sets. In summary, we have integrated a large collection of published molecular data in sarcoidosis and additional novel investigation into the revised version of the manuscript in order to address Reviewer #1's comments.*

2. Overall patient profiles are rather messy. Please clarify these queries:

- line 92 'recalcitrant' - unclear how this can be the case when patient 9 has had condition for only one year; and patient 1 was on no treatment at baseline? Please define 'recalcitrant' - is this presence of skin lesions despite how many years of treatment.

While most patients had recalcitrant disease (we define as at least several years of disease and inadequate response to prior systemic therapies), as Reviewer #1 points out, patient 9 did not have longstanding disease (~ 1 year). We also appreciate that what qualifies as "recalcitrant" might not be generally agreed up by readers and so, we have removed the word recalcitrant from the revised manuscript to avoid any confusion. We have instead added a sentence stating the average duration of disease was 13.2 years. Characteristics of disease in each patient are listed in Table 1 for the reader's reference.

*In terms of Patient 1 not being on treatment at baseline. This is because she recently refused to be treated with prednisone/methotrexate due to side effects she had previously experienced with these medications. Although she was not “on treatment” upon enrollment she had active disease (including myocardial involvement). We have added an asterix with explanation in **Figure 1a** which is annotated in the figure legend to convey this. Patient 9 (also not on treatment at baseline) had a similar situation and this has also been indicated.*

What does 'duration' in Table 1 refer to - is it duration of treatment or disease?

Table 1 has been updated to clarify that this signifies the duration of the disease.

- what is skin pattern of 'dermal' and 'subcutaneous' in table 1 - please clarify in table or text.

We have added text to clarify that this indicates the predominant histologic pattern of inflammation in the skin. This is determined with skin biopsy (but can also be inferred by clinical examination). Dermal sarcoidosis is the most common pattern, the majority of the inflammation is in the dermis, and corresponds with pink papules and plaques clinically. Subcutaneous predominant sarcoidosis is relatively less common (than dermal predominant) and is sometimes referred to as “Darrier Roussy variant”. Here, the inflammation is predominant in the subcutaneous fat and clinical lesions are erythematous, deep nodules.

Lupus pernio and massive multiple plaques are much more severe form of disease compared to occasional skin lesion as perhaps in Patient 7. Can this be expanded briefly in Table 1 given the small number of patients, and the unusual range of treatment at baseline (from no treatment in patient 1 and 7 ; to pred and MTX in patient 6). Duration of treatment also important to give a view of how resistant the disease has been to treatment.

*We have added whether or not patients had lupus pernio (LP) to **Table 1** (we appreciate this excellent suggestion). All patients had extensive cutaneous involvement, as captured by the quantitative, validated CSAMI score (e.g. **Figure 1a**) – we prefer this over massive multiple plaques which is more subjective, but if the reviewer would like to see something more specific in this regard, please let us know. To address this concern though, we have added text contextualizing what CSAMI scores of these patients mean in terms of clinical severity/management that might be more familiar to a broad audience.*

We appreciate that variability in the baseline treatment of these patients exists; however, we believe this is a realistic depiction of the wide range of treatments used in patients with longstanding/severe sarcoidosis. Early phase study of sarcoidosis is challenging in this regard in that patients risk clinical decompensation if their baseline therapy (especially prednisone) is suddenly discontinued.

*We agree that the duration of therapy is important and appreciate this suggestion. In general, any therapeutic regimen was tried for 3-6 months (or longer)- we have now noted this in the **Table 1** legend. Occasionally, the medication was discontinued sooner if limiting adverse effects were experienced. We have also stated this in the revised text. Based on this suggestion, we have also updated **Figure 1a** to depict the duration of the most recent therapeutic regimen.*

3. Briefly specify what CSAMI and TGL measure please in main text. CSAMI range at baseline is large 20-57 - please clarify what this means.

As discussed above, we have added text to clarify clinical findings contributing to score and what scores mean in terms of severity/management. We have also added text describing that patients with a CSAMI score of 10 or greater would typically be considered to have significant skin involvement and to be candidates for systemic therapy based on their skin involvement alone. We have also clarified what TLG represents to make this very clear in the main text without having to refer to the supplement.

4. Ethnicity of patients (more than skin phototype) required as the type of sarcoidosis disease in Afro-Caribbean, Caucasians etc are different

Thanks for this suggestion. We included Fitzpatrick's phototype (skin color) as a more objective measure of skin color, however, we realize this is not the only or necessarily best way to do this. We do appreciate possible differences in the natural history of disease in patients with different racial/ethnic backgrounds. Thus, we have updated **Table 1** to describe race/ ethnicity of the patients according to the latest guidelines (Flanagin et al. JAMA 2021 PMID: 34402850). We removed the Fitzpatrick phototype designations as this may create confusion.

5. Line 139 - how was lesions selected for skin biopsy, please confirm biopsy post treatment is from the same site - if not, please state why not and where skin biopsies were obtained

Thanks. This information is in the supplement, but we realize it is important and could be easily missed by the reader. We have summarized this information in the main text as requested.

6. The term 'responders' 'partial' and 'complete' responders need clarification and definition. This needs to be clearly stated from the beginning. line 142; 242 (what is 'partial' responder); 266 (what is 'best' responders - pt 2,3,4,5,8 were classed partial responders and pt 1,6,7,10 as complete/near complete responders (Fig4D) on what basis please?

Thanks for suggesting this clarification, we agree this may have been somewhat confusing as initially described. The definitions delineated below have been summarized in main text and detailed in the supplement.

Skin:

Complete response: reduction of CSAMI activity score to 0

Partial response: any reduction in CSAMI to a number greater than 0

Internal organ:

Complete/near complete response (internal organ): 98% or greater reduction in TLG (total lesional glycolysis) as assessed by PET

Partial responder (internal organ): any change in TLG that is not 98% or greater reduction

Best responders:

This designation is meant to capture changes in both skin and internal organ disease activity on therapy. Best responders were defined as patients who had **both** a complete response in the skin (CSAMI = 0 on treatment) and a complete/near complete response in internal organs (98% or greater reduction in TLG).

Why is pt 3 classified as complete responders in Fig 3J?

*For Fig 3J (now **Figure 4a**), the names of the specimens do not correspond to the patient numbers throughout the remainder of the manuscript, we apologize that this may have been confusing. We have inserted a sentence into the text to clarify that this figure panel includes data from previously published patients, hence the nomenclature change, that are cited accordingly. We have also updated the Figure legend and text to call out a table summarizing sample designations (**Supplementary Table 4**; previously Table S3). We have also updated the nomenclature in this Figure panel to provide further clarity.*

6. Line 146 - 8 patients had pulmonary involvement - Table 1 states only 4 had lung involvement - did you count thoracic lymphadenopathy as lung involvement? If so, please state as this is debatable; most clinicians would view pulmonary involvement as presence of interstitial lung disease.

*Thanks for prompting this important clarification. We have added a supplemental table (**Supplementary Table 1**) which includes more extensive detail about the pulmonary disease in these patients including Scadding stages and results of pulmonary function testing (PFTs). To answer the question: 6 patients were Scadding stage 2 (e.g. parenchymal infiltration) and 3 patients were Scadding stage 1 (thoracic lymphadenopathy only). To also make this clear in the main text, we added the Scadding stage of each patient to **Figure 1a** (previously Figure 2A) and **Table 1** (as well as definitions of the Scadding stages to the Table 1 legend) so this information is readily available.*

*In terms of the apparent discrepancy 8 patients vs 4 patients, we have also clarified in the **Table 1** legend that Scadding stage depicted represents the highest documented Scadding stage in the past. Overall, the status of pulmonary involvement for each patient is clearer in the revised manuscript.*

7. table 1 please state what is n/a - not assessed? not available?

Thanks. n/a has been replaced with "none known" when referring to other sites of involvement. We realize the use of both "n/a" and "no" within this table was confusing.

8. Figure 1E - unclear how dactylitis is visible on picture (!!) and not mentioned in Table 1. the only change visible in Fig 1E appears to be in toe nails?

*Figure 1E is now **Supplementary Fig. 1**. Dactylitis refers to the erythema and infiltration (larger circumference) of many toes (R1, L1, L5 toes in the clinical pictures). On the top the toes are a deep red and more swollen. On the bottom the toes are much less erythematous and also less swollen. We realize it may have been difficult to appreciate this given the small size of the picture. As part of the revision of the manuscript, this panel was moved to the supplemental section and is now larger and we hope easier to appreciate.*

*Symptomatically the patient can now walk without pain and this is a truly a significant improvement for him. For context, the R3 toe was amputated remotely (missing in the clinical image) because it was so uncomfortable to the patient. We have also added the presence of dactylitis to **Table 1** as requested.*

In terms of the nail changes, sarcoidosis can also involve the nail matrix leading to nail plate dystrophy; the patient did have some improvement in nail dystrophy on therapy as well, but this

was less bothersome to him (an example of improvement in nail dystrophy in another patient was introduced into the revision, **Supplementary Fig. 2**).

8. Figure 1 G - pls specify where the biopsy is from (cf point 5 above)

Figure 1G is now **Figure 1e**. This clarification has been added to the legend. We have also provided additional clarity in the methods (main text) on how biopsy sites were selected in general.

9. Figure 2A. Where if Pt 1 on graph pls?

Figure 2A is now **Figure 1c**. All patients with systemic involvement (n=9) received a whole body PET-CT plus a dedicated cardiac PET-CT study at baseline and after 6 months of treatment (one post-treatment studies was uninterpretable due to noncompliance with the dietary prep). Pt1 was the only patient that had myocardial disease. This myocardial disease activity in Pt 1 was below the stringent threshold used to quantify the whole-body PET-CTs. Thus, in this patient the TLG value for the whole-body PET-CT was 0 at baseline and at 6 months (eg **Figure 1c**). Rather, the myocardial activity was better quantified using the dedicated cardiac scan. The results from the dedicated cardiac scan are shown in **Supplementary Fig. 4** (originally Figure 2F). We agree this was confusing and have added additional text to clarify this in many areas. We have also added an entire section to the Supplement which discusses this issue in great detail.

10. Figure 3A and scRNAseq studies:

-this is the main strength of the paper but limitations need to be stated. Only 3 patients in a heterogenous disease and patient numbers (1,2,5) need to be stated upfront in the text (not suppl data) and the fact that one is on no treatment and the other two on MTX and Pred (for ?years) need to be specified as this will likely impact on gene expression profile.

We appreciate this feedback and have made the following updates to address this:

1. As discussed above, Figures 1 and 2 (clinical data) have been condensed into a single main figure (**Figure 1**) so that 3 main figures could be devoted to the molecular analysis including the scRNAseq analyses
2. We have added explicit statement that two of the patients were on treatment when the scRNAseq samples were collected and agree the gene expression patterns could be altered by the treatment. We do note, however, here and in the revised text, that despite baseline treatment (which had been stable / recently unchanged), granulomas were still present in the skin and we would argue that thus the core signals driving granuloma maintenance were still intact. Indeed, we found similar molecular changes by scRNAseq in all three of these patients irrespective of baseline treatment.
3. As discussed in relationship to Reviewer #3's comments, we have also improved the depth/rigor of the scRNAseq analyses in general (see below).
4. As discussed in detail above related to comment #1 from Reviewer #1, we have addressed clinical heterogeneity, small sample size, and skin-biased analysis by including : A) analysis of pulmonary sarcoidosis (BAL) scRNAseq data vs controls, B) analysis of bulk gene expression data from a large cohort of pulmonary sarcoidosis (BAL) patients with diverse phenotypes vs controls, C) analysis of FACS-purified myeloid populations from pulmonary sarcoidosis versus controls, D) analysis of bulk RNAseq data from a large cohort of patients with cutaneous sarcoidosis, and E) analysis

of cytokine expression patterns in additional pulmonary and cutaneous sarcoidosis tissue biopsies using RNA in situ hybridization staining. We integrate and describe these multiple complementary datasets using orthogonal approaches to validate the key findings of the skin scRNAseq data from these 3 sarcoidosis patients and we believe this speaks to the generalizability of the findings.

- Figure 3 B and E - confusing. What is "SAR other" please? It seems also that cluster 2,7 and 12 are over represented differentially in all three patients?

Figure 3B and 3E are now in Figure 2a. We have reorganized our scRNAseq data presentation substantially in the revision, including devoting additional main figure space to improve clarity. We have renamed the T cell and macrophage subset designations in the revised manuscript in order to provide additional clarity. We have also included additional, new description of characteristics of the "SAR other" T cell population (now called CD4+ SAR-2) and find that these cells express markers of a recirculating population of T cells (e.g. express lymphoid homing markers) (Supplementary Fig. 6). Similar populations have been observed in scRNAseq data from other inflammatory skin disorders (Liu et al. J All Clin Immunol 2021, PMID: 33309739). This is discussed in the revised text.

We agree that clusters 2,7,12 are highly enriched in individual patients. (We demonstrate that these populations represent key CD4+ T cell clusters in sarcoidosis). As discussed above, it is indeed expected that there is some degree of heterogeneity from patient to patient. Additionally, as discussed above, as two patients were taking other medications when these samples were collected, variability is not surprising. Given these considerations, we find this observation to be reassuring and to support our hypothesis; that is: a population of CD4+ T cells characterized by evidence of chronic antigen exposure, a Trm phenotype, and increased production of IFNG and CSF2 (GM-CSF), in addition to specific chemokines, is present in each patient.

Fig 3C - how were these transcripts selected? pls specify. Suggest move Figure S3 to main figure to address this.

Figure 3C is now Figure 2d. Thanks for this point, we have indeed moved much of Figure S3 into the main section (now Figure 2c and 2e). To answer the questions, these genes are a curated gene list that includes the most differentially expressed genes (best visualized with a volcano plot, now Figure 2c) plus selected canonical T cell markers. Similar changes have been made to clarify analogous plots for myeloid cell populations (e.g Figure 2f-j).

11. Fig 4C - left lower quadrant probably just as important as right upper quadrant.

Figure 4C is now Figure 4d. We agree. Accordingly, we have labeled key immune genes in the left lower quadrant. We have also done this for prior Figure 4A and 4B, now Figure 4b and 4c respectively.

Reviewer #2 (Remarks to the Author):

The authors have submitted a report on an open label study of Tofacitinib in sarcoidosis, accompanied by supporting laboratory investigations to provide mechanistic explanation for the drug.

The clinical study is well described. The authors sought local approvals to undertake the work. There was sponsorship from the company that manufactures Tofacitinib.

The 10 patients are representative of a severe sarcoid cohort. Table 1 is important in understanding the disease patterns. One question - I was unsure on the difference between 'no' and 'n/a' in the Other column.

This point was also raised by Reviewer #1 and has been addressed/clarified as above. We apologize that this was not clearer in the original version.

The supplementary data are excellent, and have answered all the questions I had regarding the population, outcomes and adverse event profiles.

The PET images and skin photographs provide useful context, and the images show very typical sarcoid disease manifestations.

The paired laboratory data follow a logical flow. I have looked through each figure, and the plots are easily interpretable, and all appear genuine.

The cytokine profiling is consistent with the authors' hypothesis regards the IFN suppressive effects of Tofacitinib.

Overall, the manuscript is a very compelling argument to undertake further study of this therapeutic strategy in sarcoid.

We thank Reviewer #2 very much for the positive feedback.

Reviewer #3 (Remarks to the Author):

This report describes an open label trial where 10 patients with sarcoid recalcitrant to other treatments were with 5 mg of tofacitinib twice a day, without stopping other treatments or washout. Response was assessed using a skin-focused clinical activity and morphology score at 6 months and also by TLG assessed by PET. Greater than 80% CSAMI improvement was observed, with 6 patients showing a complete response – this number was 3/10 for TLG:PET.

scRNAseq analyses of affected tissue before and after treatment pointed to inhibition of CD4+ T cell interferon production as a cause for improvement. Specifically, CD4+ T cell clusters were identified for 3 sarcoid patient biopsies not found in normal controls, and in one such cluster, IFNG was the most upregulated cytokine, compared to modest TNF and minimal IL-17 expression. IFNG bias was clear in the overall scRNAseq analysis for sarcoid samples and normalized with tofacitinib treatment. In the APC analysis, the most interesting data likely regarded a population of cDC1-like cells showing transcript expression of IL12 in the sarcoid samples.

A series of serum sarcoid inflammatory markers were also found in correlation with disease severity and response to jakinib, including CXCL9, CXCL10, and CXCL11.

It was challenging for this reviewer that the title and first half the abstract read like a clinical study, and that the second half of the abstract veers into molecular analysis. It is really only this second half that has the potential for novelty, but as described below, I have serious questions about the depth and rigor of this deeper molecular investigation.

We appreciate this feedback. In order to address the concerns in the above paragraph, we have taken the following steps:

1. We have significantly restructured the manuscript to allow us to delve deeper into the prior data and also to add additional, new molecular analyses. The molecular work now constitutes 3 main figures, instead of 2 as in the original submission. The new molecular analyses are discussed in great detail above in response to Reviewer #1's comments.
2. We have also retitled the manuscript and updated the abstract to try to improve the translational message.
3. Additional issues raised by Reviewer #3 are discussed with more specificity below and also help to address this general concern.

I think the authors are aware that the real question is not whether these findings are clinically significant in a broad sense. (They are.) Rather, the question regarding whether this paper belongs at NatureCommunications centers on whether something distinctly new is brought to the table. There has been a recent run of valuable studies on treatment of sarcoid using jakinibs including tofacitinib, none as prominent as one by these authors in NEJM.

We thank reviewer 3 for raising these many important points and are glad to have the opportunity to discuss the potential significance of the clinical portion of this work further. In terms of lack of novelty for the clinical portion, we respectfully disagree, in that, although this is not the first report of this treatment strategy in this disease, **prior reports (including our NEJM report) are either case reports of single patients or very small retrospective case series. These types of reports have a significant potential bias in selecting for patients that responded positively to therapy.** This is particularly true in a disease with such clinical heterogeneity (as noted by Reviewer #1).

The present clinical study is a **prospective evaluation** of this approach, which is an important distinction from the prior work and a natural progression of the work. Often, in this setting, efficacy is less than might have been suggested by retrospective case reports. Along these lines, **we believe that the magnitude of the clinical response in a prospective setting is strikingly robust (10/10 patients responded and improved to a degree not previously achieved with the prior medications, including standard clinical approaches).** Further, **this is a disease without FDA approved therapies (other than prednisone) that commonly affects black/African American patients and is in desperate need of safer more effective therapies.** The fact that the magnitude of response was so high, in our opinion, is unprecedented in sarcoidosis, and in-and-of itself very significant. This prospective evaluation of 10 patients sets the stage for randomized studies in a way in which small retrospective case reports do not. As discussed below, we believe that the molecular work supplements this and provides a clear path forward for larger clinical trials (in terms of response patterns, outcomes, and predictions about which JAK proteins(s) might be key to inhibit).

In that NEJM article, not only was substantial improvement of sarcoidosis in response to tofacitinib noted, but the bulk RNAseq results showed “While the patient was taking tofacitinib, there was down-regulation of the messenger RNA (mRNA) in the JAK-STAT-dependent pathways (interferon- γ and interleukin-6) as well as of mRNA in pathways that are not directly regulated by JAK-STAT (TNF α and mTORC1).”

Regarding pre-and post-treatment, the NEJM paper notes: Gene-set enrichment analysis showed activation of interferon- γ and tumor necrosis factor (TNF) α signaling as well as interleukin-6–STAT3 and mammalian target of rapamycin complex 1 (mTORC1) signaling in the pretreatment skin-lesion samples; in contrast, in skin-lesion samples obtained during treatment and in normal skin samples, these pathways did not appear to be activated.”

The cytokine list is longer in this study, but the precision is not much greater.

We understand Reviewer #3’s points here. In the revised manuscript, we have expanded the molecular work to a much larger group of patients using new approaches and have also expanded the molecular work beyond the skin (including lung and blood). The new molecular work is detailed above where addressing Reviewer #1’s comments (especially comment #1). Not only do we analyze a much larger number of patients, we also provide further precision on the immunologic signals well beyond that provided through prior work. In particular, through our single cell studies, we identify central producer and receiver cell types of key cytokine signals in sarcoidosis and describe how they correlated with clinical improvement. We find that CD4+ T cells producing IFNG and GM-CSF are key drivers of disease. We find that IL-6 and IL-15 are also important and primarily produced by stromal cells (fibroblasts and endothelial cells). We also describe a population of cDC1s that produce IL-12 in sarcoidosis (discussed further below). We also validate these and other cytokine expression patterns within both skin and lung tissue.

In sum, although these data support some of our prior hypotheses (eg IFNG and IL6 are important); the precision and scope of the present work, in our opinion, is a significant advance over prior work.

The single-cell analysis here is simplistic, focusing on CD4 and CD8 categories. Problematically, there are virtually no CD8 cells in the controls and it may be that all control CD4 cells are clustered separately into a cell-type called “Ctrl”? It would be far more reassuring to recognize different resident memory populations, levels of exhaustion in CD8 cells etc.

We appreciate this feedback and have updated our analysis of the single cell data to make it more comprehensive. In particular we have focused on the following:

1. **For CD4+ T cells**, we have looked at additional canonical markers of T cell activation, exhaustion, functional polarization, and T resident memory (Trm) markers to help address this concern. This data is presented in **Supplementary Fig. 6**. We do find that the pathogenic T cells in sarcoidosis express T resident memory markers. [The T cell clusters, including resident memory T cells, that we observed in the normal skin samples are consistent with other prior scRNAseq studies comparing normal skin and inflammatory disorders (e.g. Gellatly et al. *Sci Transl Med* 2021, PMID 34516831)]. In particular, multiple distinct clusters of FOXP3-CD4+ T cell in normal skin is not necessarily expected. We have also looked at and characterized other CD4+ T cell populations in sarcoidosis samples and find they express lymphoid homing markers (this issue is discussed above in response to Reviewer #1). As discussed above, we also look at a pulmonary sarcoidosis scRNA-seq dataset to study CD4+ T cells in this setting as well. We discuss these multiple additional considerations in the revised manuscript.

2. **For CD8+ T cells**, we agree that there were very few CD8+ T cells in normal skin (only 71 cells total). This is expected in that normal skin, CD4+ T cells typically outnumber CD8+ T cells (e.g. Koguch-Yoshioka et al. 2921, *Commun Biol*, PMID 33398080). For this reason, our ability to detect significant differences in gene expression from CD8+ T cells was limited. Below, for reference, are the most differentially expressed genes when comparing CD8+ T cells in the sarcoidosis libraries to the control libraries; it is difficult to draw and certain biologic conclusions because the cells look quite similar. This is consistent with the clustering which did not show distinct clustering of CD8+ T cells from sarcoidosis versus normal controls. We discuss this observation (and its limitations) but do not present this data in the revised version of the manuscript, if it was preferred we include this, we certainly would be happy to. [To further contextualize this, we believe the CD8+ T cells may be less important in sarcoidosis based on a number of prior observations including: increased cytokine production by CD4+ T cells, expansion of CD4+ T cell clones in sarcoidosis, increased number of CD4+ T cells in sarcoidosis (relative to CD8+) and association of sarcoidosis with genetic variation at MHC class II loci (reviewed in Drent et al. *New Engl J Med*, PMID: 34496176).]

Upregulated in sarcoidosis CD8+ T cells			Upregulated in control CD8+ T cells		
Gene	avg_log2FC	adj p value	Gene	avg_log2FC	adj p value
RPS26	2.25857264	2.43E-32	IFITM1	-1.0177797	6.51E-09
XIST	1.78461001	1.74E-10	SQSTM1	-1.0275697	9.13E-05
COTL1	0.94349301	0.00831749	FOSL2	-1.0373191	9.01E-05
JUNB	0.80794773	0.02735682	ATP2B1	-1.0497751	0.00031693
RPS4X	0.58973087	4.75E-07	AC058791.1	-1.0763204	0.00517113
RPLP1	0.52814065	8.82E-07	LMNA	-1.1761101	0.00460182
RPS19	0.52796174	8.29E-05	ANKRD28	-1.1817001	0.00050222
RPL9	0.42728238	0.04799502	DDX3Y	-1.2781503	2.59E-67
B2M	0.41457122	4.79E-08	RPS4Y1	-1.307501	6.40E-48
RPL28	0.38259455	0.02838346	MTRNR2L1	-1.3808733	2.09E-65
			CRIP1	-1.5118915	5.23E-11
			MTRNR2L12	-1.6260484	2.49E-16
			XCL2	-1.6699086	0.04108573
			GNLY	-1.9830319	3.58E-13

3. **For macrophages**, we have added additional description and discussion of macrophage heterogeneity in sarcoidosis (see **Supplementary Fig. S7**).
4. **For DCs** (which this reviewer agrees is interesting), we have increased study of the IL12B producing cDC1s that we find to be enriched in sarcoidosis, with validation of this finding in two new additional data sets (**Figure 3d, S10**).
5. **For fibroblasts**, we have expanded analysis of this cell type and moved some of the data into the main figures (see **Figure 3f**) in addition to new data presented in **Supplementary Fig. 12**.

I came away understanding for sure that IFNG and CSF2 are up in CD4 cells in sarcoid lesions and are normalized during treatment and correlate with response. But what does this bring to a casual or expert audience vs. the previously published papers? I think there is some novelty in this study, on a level well served in JID or JACI.

We also appreciate this feedback and during our revision we took a step back and reflected greatly on this particular point: about what is being brought to both casual and expert audiences.

For casual audience:

We took a broader approach to the immunologic changes in sarcoidosis as part of the revision. In particular, we evaluated other competing hypotheses around basic immune polarization in sarcoidosis. In particular, we refocus the manuscript in some areas to address the question of whether Type 1 (e.g. Th1), Type 2 (e.g. Th2), or Type 17 (e.g Th17) immune responses predominate in sarcoidosis. In a very recent New England Journal of Medicine review article on sarcoidosis (Drent et al. New Engl J Med, PMID: 34496176), there is still discussion/debate over whether or not Th17 or Th2 cytokines are involved in disease pathogenesis.

We took a very structured approach to this question, as exemplified by our analysis of multiple different data sets for canonical markers of Type 1 vs Type 17 vs Type 2 immune polarization and find very little support for Type 17 and Type 2; this helps to refocus our findings around Type 1 cytokines in a way we believe will be important to a more casual audience.

For expert audiences:

We believe our model of cell-cell communication including identification and validation of core cytokines in this disease will be of great interest. In particular, delineation of these cytokine networks will be essential to guiding selection of an optimal JAK inhibitor(s) (of which there are multiple with varying specificity) for the next phase of clinical evaluation of this approach in sarcoidosis.

Given the breadth of revisions and for all of the reasons discussed above, we believe we now present a significantly improved manuscript which we hope will be considered seriously at Nature Communications, as opposed to another journal.

REVIEWERS' COMMENTS

Reviewer #1 (Remarks to the Author):

This paper is massively improved but unfortunately does have substantial addition which were actually not requested by any of the reviewers.

Nevertheless, I have read and worked through all these data very carefully; in part as I am familiar with the ERJ (Liao) and GRADS data.

The messaging is better but greatly diluted by the new data which is inferior to the old ones that were reviewed and improved. The title is much better but not right as the authors have not shown that inhibition of T1 immunity with Tofa INDUCED improvement. To do this will require all the requisite controls which is not possible in this study. Suggest change this to 'Inhibition of T1... ASSOCIATES with marked improvement...'

Overall, the new addition drags down the paper (and please note that these were not requested by any reviewers) and also does not add much to the field nor the paper. There is v good evidence that type 1 immunity is the core immune abnormality in sarcoidosis.

Figure 3d needs some more work -positive controls, high magnification picture of histiocytes for example. Figure 3e and 3g are unnecessary and invites discomfit as it is removed from the point of the paper, and as it is in silico, it will need validation. As such the authors have raised hypothesis rather than supported their premise.

Figures 12,4 are all greatly improved and on their own should be interesting and sufficient data

Line 54

'conserved cytokine mediator of macrophage activation'.. this is not clear and I am not sure is shown in the paper

Line 127

please point to Suppl Table here of the patients involved, pls specify if these are new patients and provide identity of these page with ref to Fig 1a.

Line 251

Here 6 post Tofa treated skin biopsies were compered to skin from 6 healthy controls ; and another 15 sarcoidosis and 5 controls? Fig 3a is not clear - which samples were compared to which samples and both Tofa treated and published untreated bulk RNA seq data were compared to their healthy controls - but 'predicted upstream regulators' were shown instead of DEGs. The plot refers to enrichment which is usually used for gene sets rather than single genes - please clarify this plot. How were predictors derived - pls mention a method/ref.

I don't think it is relevant to have all the data on pulmonary sarcoidosis in here as this is a skin-based indication for treatment and it is difficult to extrapolate to the lungs. it detracts from what the authors have already - pre and post skin immunology after Tofa treatment which is the novel part here.

line 328 - not clear what 'conservation of molecular drivers in sarcoidosis' means. Can you clarify.

Reviewer #2 (Remarks to the Author):

I am satisfied with the revised manuscript.

Reviewer #3 (Remarks to the Author):

The revision regarding differentiation from the prior work has substantially addressed this Reviewer's comments and is objectively improved. The single cell analysis in particular is far clearer and acceptable for publication.

Reviewer #1 (Remarks to the Author):

This paper is massively improved but unfortunately does have substantial addition which were actually not requested by any of the reviewers.

Nevertheless, I have read and worked through all these data very carefully; in part as I am familiar with the ERJ (Liao) and GRADS data.

The messaging is better but greatly diluted by the new data which is inferior to the old ones that were reviewed and improved. The title is much better but not right as the authors have not shown that inhibition of T1 immunity with Tofa INDUCED improvement. To do this will require all the requisite controls which is not possible in this study. Suggest change this to 'Inhibition of T1... ASSOCIATES with marked improvement...'

Thank you for the feedback. We have changed the wording of the title from “includes” to “is associated with”.

Overall, the new addition drags down the paper (and please note that these were not requested by any reviewers) and also does not add much to the field nor the paper. There is v good evidence that type 1 immunity is the core immune abnormality in sarcoidosis.

Per discussion with the Editor, it is preferred to keep this data in the manuscript as we anticipate it will make the work of broader interest.

Figure 3d needs some more work -positive controls, high magnification picture of histiocytes for example.

We have included higher magnification insets of representative positive cells as requested. Representative examples of positive controls are also included.

This data is now found in **Figure 7a-b**.

Figure 3e and 3g are unnecessary and invites discomfit as it is removed from the point of the paper, and as it is in silico, it will need validation. As such the authors have raised hypothesis rather than supported their premise.

We would defer to the Editor on how best to handle this comment and if additional changes are needed. From our point of view, data in original Figure 3e is quantification of the data shown in Figure 3d, which is discussed above and important for our conclusions. It seems helpful to include these data. These data are now in **Figure 7a-b**.

Original Figure 3g shows the results of a receptor-ligand interaction analysis (Cellphone DB). From our perspective, this data is useful and helps to reinforce the key findings / interpretation of the scRNA-seq data analysis in a relatively unbiased fashion. Thus, we would prefer to keep this figure panel in as well, but if the Editor prefers we remove this, we would be happy to. We attempted to address this concern by updating the text according the reviewers comment to point out that this is an “in silico” analysis. This data is now found in **Figure 8b**.

Figures 12,4 are all greatly improved and on their own should be interesting and sufficient data

Line 54

'conserved cytokine mediator of macrophage activation'.. this is not clear and I am not sure is shown in the paper

We have changed “conserved” to “central”, we hope this will address this concern.

Line 127

please point to Suppl Table here of the patients involved, pls specify if these are new patients and provide identity of these page with ref to Fig 1a.

We have referenced Supplementary Table 4 here as requested. Thanks for pointing out that clarification was necessary here. We have also added to the text so that it is also clear which samples are being studied with regard to **Figure 1a** without having to reference the Supplementary Table.

Line 251

Here 6 post Tofa treated skin biopsies were compered to skin from 6 healthy controls ; and another 15 sarcoidosis and 5 controls? Fig 3a is not clear - which samples were compared to which samples and both Tofa treated and published untreated bulk RNA seq data were compared to their healthy controls - but 'predicted upstream regulators' were shown instead of DEGs. The plot refers to enrichment which is usually used for gene sets rather than single genes - please clarify this plot. How were predictors derived - pls mention a method/ref.

We appreciate this could be confusing. We have changed the axis labels in original Figure 3a (now **Figure 6a**) to help make this clearer. We have also updated the description of this data in the main text to improve clarity. We think these changes make this much clearer and will avoid confusion. The predicted upstream regulator module of IPA was used. This is a commercially available pathway analysis tool described in the supplemental methods section. The content version is also listed in the supplemental methods section. We have also now listed the content version for IPA in the figure legend so that the reader can see that these are coming from the IPA database/analysis platform.

I don't think it is relevant to have all the data on pulmonary sarcoidosis in here as this is a skin-based indication for treatment and it is difficult to extrapolate to the lungs. it detracts from what the authors have already - pre and post skin immunology after Tofa treatment which is the novel part here.

As per discussion with the Editor, we prefer to keep this data.

line 328 - not clear what 'conservation of molecular drivers in sarcoidosis' means. Can you clarify.

We meant that the drivers appeared similar among patients and among disease sites. We have changed “conservation” to “similarity in”.

Reviewer #2 (Remarks to the Author):

I am satisfied with the revised manuscript.

Reviewer #3 (Remarks to the Author):

The revision regarding differentiation from the prior work has substantially addressed this Reviewer's comments and is objectively improved. The single cell analysis in particular is far clearer and acceptable for publication.